# Periodic corner holes on the Si(111)-7×7 surface can trap silver atoms

Jacek R. Osiecki [1,2✉], Shozo Suto [2✉] & Arunabhiram Chutia [3✉]

Advancement in nanotechnology to a large extent depends on the ability to manipulate materials at the atomistic level, including positioning single atoms on the active sites of the surfaces of interest, promoting strong chemical bonding. Here, we report a long-time confinement of a single Ag atom inside a corner hole (CH) of the technologically relevant Si(111)-7×7 surface, which has comparable size as a fullerene $C_{60}$ molecule with a single dangling bond at the bottom center. Experiments reveal that a set of 17 Ag atoms stays entrapped in the CH for the entire duration of experiment, 4 days and 7 h. Warming up the surface to about 150 °C degrees forces the Ag atoms out of the CH within a few minutes. The processes of entrapment and diffusion are temperature dependent. Theoretical calculations based on density functional theory support the experimental results confirming the highest adsorption energy at the CH for the Ag atom, and suggest that other elements such as Li, Na, Cu, Au, F and I may display similar behavior. The capability of atomic manipulation at room temperature makes this effect particularly attractive for building single atom devices and possibly developing new engineering and nano-manufacturing methods.

[1] MAX IV Laboratory, Lund University, SE22100 Lund, Sweden. [2] Department of Physics, Tohoku University, Sendai 980-8578, Japan. [3] School of Chemistry, University of Lincoln, Brayford Pool, Lincoln LN6 7TS, United Kingdom. ✉email: jacek.osiecki@maxiv.lu.se; shozo.suto@tohoku.ac.jp; achutia@lincoln.ac.uk

About sixty years have passed since the famous talk by Richard Feynman entitled, "There is plenty room at the bottom"[1], which envisioned the future of nanotechnology as a possibility to manipulate, control and utilize matter at the level of single atom in the way we want i.e., atom by atom. Certainly, the age of nanotechnology has arrived to some extent as Feynman predicted but still, we have a long way to go to comprehend the atomic world and to use the knowledge for the good and prosperity of mankind.

Building a device based on a single atomic species is one of the ultimate goals in nanotechnology[2–4]. The realization of such a device is technologically non-trivial and still poses a challenge. The main reason for the difficulty in realization of such an endeavor is that the atom in such a case should be confined in a reasonably small space, possibly buried and ideally be locked in one desirable atomic position. Several possibilities may exist e.g., adhering the atom on the adsorption sites of a surface at very low temperature or through chemical bonding (covalent bonding) or inserting an atom forcefully inside a crystal as an interstitial through ion implantation. The last two methods might not require low temperature. The method of adhesion of a single atom at one place on a surface using low temperature might be most widely and readily used. However, it is most prone to instabilities, technological difficulties, impracticality and require substantial investments. Therefore, there is a need for a method of atomic entrapment on a specific site that can be applied to several elements, which can be easily used, performed at room temperature, and can be cooled down if treatment and technology requires.

Adsorption of a single atom or a molecule on the silicon surface at one place can be achieved through a dangling bond (unpaired electron) at room temperature, which may require controlled conditions such as ultra-high vacuum (UHV) or a specific gas atmosphere. For instance, site-specific chemisorbed and physisorbed molecules on the Si(111)-7×7 surface have been explored by Lock et al.[5]. In another study, controlled atomic site adsorption and positioning have been achieved for example, on hydrogen-terminated Si surface. The hydrogenated surface has all dangling bonds saturated with hydrogen[6] and by selectively removing the hydrogen atoms and by exposing to a specific gas e.g., phosphine gas can result in phosphorus atoms in desirable positions after heat treatment[3].

The silicon crystal has been subjected to tremendous scientific scrutiny especially, the Si(111) surface has been the subject of many experiments and theoretical work. Interestingly, the Si(111) surface reconstructs into a 7×7 structure which has a relatively large unit cell and it can be further divided into two similar half subunits called as half unit cells (HUCs). Two topmost surface double layers of atoms are taking part in the reconstruction of 7×7 most actively[7]. The adatoms and the atoms in the second layer can be observed using STM. The 7×7 structure also has a characteristic and quite unique corner hole, which is a structural hole, and looks like a hollow feature in the STM images[8]. Even though the corner hole is relatively small, its diameter is comparable to the $C_{60}$ molecule[9]. The internal structure of the corner hole differs most noticeably from $C_{60}$ by having one unpaired electron in the form of a dangling bond of Si atom at the bottom center. There is an interest and a lot of work that has been devoted to encapsulating atoms inside $C_{60}$ molecules already[10]. It seems that the corner holes on the Si(111)-7×7 in a way offer a similar possibility but without full encapsulation and they are already positioned directly onto the surface forming a periodic array. The corner hole of the Si(111)-7×7 is predicted to be the most reactive of all sites of this surface. According to the theory of local softness and charge capacity, the corner hole is unique in exhibiting a strongly active site for electrophilic reactants as well as exhibiting some reactivity for nucleophilic reactants compared to other possible sites on HUCs[11]. The initial adsorption of single atoms on the Si(111)-7×7 surface was investigated experimentally with a handful types of elements, which include Ag, Au, Pb, Cu, and Yt.[12–16]. All the above findings show that atoms adsorb inside half unit cells and atomic diffusion is constrained between HUCs due to the 7×7 reconstruction. It was also found that atomic elements on the 7×7 surface can be manipulated using STM, which can be performed even at room temperature (RT)[17,18].

Despite a broad interest on the Si(111) reconstructed surface for some reason chemical characteristics of one adsorption site i.e., corner hole has not been explored enough. The site by itself seems to be attractive because of its size, position, chemical character, and periodicity. Therefore, foreign atoms in the corner holes need to be explored more in detail, which is the aim of this study.

Here we show that the CH which is a structural round depression in the unit cell of the Si(111)-7×7 surface can accommodate single Ag atom and can trap it at room temperature. This finding is supported by the experimental and theoretical calculations. The silver atom can diffuse out of the corner hole at a high temperature. The ability to trap the Ag atom might be very important for the controlled placement of atoms at stable adsorption sites and building stable single-atom devices on a semiconducting surface.

## Results

**The Si(111)-7×7 surface.** The Si(111)-7×7 surface is resolved in real space using a scanning tunneling microscope (STM). Relatively large unit cell having a diamond shape consists of two similar triangular subunits called half unit cells (HUCs) which is shown in Fig. 1(a) and (b). In a filled state image, we can distinguish between faulted and unfaulted half unit cells, which is shown in Fig. 1(b) and they are denoted by letter F and U respectively. Each half unit cell consists of six protrusions arranged into triangular shapes. Faulted halves have an appearance of being a little brighter than unfaulted halves in filled state images. At each corner of every unit cell there are four corner holes (CHs) that look like dark hollow features. Each corner hole is surrounded by six adatoms and alternately around the corner three of them appear brighter than the other three. One of the corner holes is denoted in Fig. 1(b) as "empty CH".

**The Ag atoms on the Si(111)-7×7 and the STM images.** It is possible to prepare clean Si(111)-7×7 surfaces with low defect density (see. supplementary figure 1 in Supplementary Data 2) and on these clean surfaces we deposited silver in the range of thousands of the monolayer (ML) from an e-beam evaporator (monolayer is $7.81×10^{14}$ atoms/cm²). This amount of silver gives roughly every Ag atom per 10 to 20 HUC. At that small surface density of silver atoms, the room temperature STM images reveal some HUCs as brighter than the other as shown in Fig. 1(a) and for clarity it is shown in the inset figure, Fig. 1(a). The silver atoms are trapped within the boundaries of HUC, which results in the change of its appearance. The appearance of the HUCs with silver atoms is explained by the process of fast intra-cell diffusion of a silver atom inside HUC. The atomic movement is much faster than raster scanning of the STM microscope and what we see is an average result. Cooling to liquid helium temperatures (about 4 K) stops the atom from intrabasin hopping inside HUCs and the atoms can be imaged at one position inside a relatively large HUC (area about 3.2 nm²)[12].

Careful visual inspection of the STM images at the initial state of the surface after deposition with Ag atoms reveals a changed

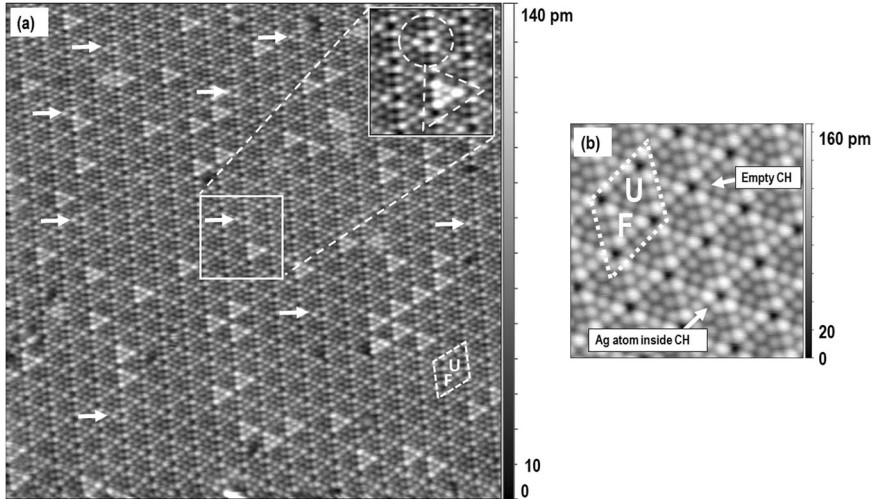

**Fig. 1 Si(111)-7×7 surface with Ag atoms.** (**a**) Filled states STM image of the Si(111)-7×7 surface with Ag atoms obtained at -1.8 V, and 177 nA (size 50 × 50 nm²). The basic unit cell is denoted in the image with the diamond shape outline. The Ag atoms in the corner holes (CHs) are pointed with the arrows and can be recognized with the surrounding six top adatoms that belong to six adjacent unit cells and those adatoms look brighter than those around the empty corner holes. The inset in the right top side clearly shows CH with Ag atom denoted with the circle and arrow and also Ag inside FHUC outlined with the triangle. (**b**) High-resolution scanning tunneling microscope (STM) image (filled states, −2.0 V, 153 nA, 12 × 12 nm²) with two corner holes denoted by the arrows, one pointing to the corner hole with 1 Ag atom and another pointing to the corner hole without Ag atom. Original files in (a) and (b) are provided in Supplementary Data 2: m17_ori.par, m17_ori.tf0, m17_ori.tb0.

look of few corner holes per unit area. Six adatoms that are around these different corner holes appear to be brighter than others. It turns out that the Ag atoms sitting inside make the adatoms around those holes appear brighter as presented in Fig. 1(b). It is observable already in the first image obtained after the deposition. The bottom of the corner hole with the Ag atom is only five picometers higher than the empty one. In case of adatoms surrounding CH they are also higher and the magnitude depends on the adatom (for high difference see supplementary figure 2). The appearance of the corner hole with Ag inside and surrounding adatoms has some brightness variation depending on the state of the tip and tip bias.

In case of an unfavorable condition of the tip, especially too low bias voltage, the corner hole looks closer to an empty one. In our experiments the state of the tip (i.e., the chemical composition, and atomic arrangement) was undefined, but it is presently unavoidable to operate the STM without such knowledge of the state of the STM tip. We note that the change of the state of the tip to preferential condition is frequently and easily done with voltage pulsing and tip bias voltage adjustment (typically from about - 0.8 V till −2.0 V) till one can see the proper quality and enough contrast to visually discern the presence of the Ag in CH.

We have analyzed the height difference that makes the adatom look brighter next to the CH with Ag inside with respect to the nearby corresponding adatoms next to the CHs without the Ag atoms. The analysis of the measurement distribution is shown in supplementary figure 3. The height difference distribution is a normal Gauss distribution indicating the existence of one height value for Ag with a standard deviation related to measurement error supporting the existence of CH with Ag atom.

Almost immediate occupation of the corner hole by a single Ag atom just after deposition suggests that atoms impinging on the surface can find a way to the corner holes from the vapor in the process of direct site landing or through relaxation and diffusion. At a relatively small amount of Ag atoms (about 1 atom per 10 HUCs) the ratio of the atoms being inside the corner hole to all atoms arriving at the surface was 4.67 % (for all 364 Ag atoms 17 were inside CH). It is slightly higher than the ratio of the area of

the corner hole to the whole area of the unit cell (close to 3%) which might support the existence of both relaxation processes on arrival and direct corner hole adsorption mechanisms.

At RT, the Ag atoms are contained in HUCs but also diffuse and jump between the faulted and the unfaulted HUCs. The process of inter-cell jumping is shown in Fig. 2(a) and (b), which are two consecutive images of the same area. The Ag atom occupying faulted half unit cell (FHUC) changes the HUC and jumps to one of the nearest unfaulted half unit cell (UHUC). This process at RT is slow and in the order of minutes or hours depending on the direction of the jump[11].

Here we find that the Ag displays quite a rare jump into a corner hole with the rate lower than intra-cell hopping from FHUC to UHUC. Figures 2b, c present the jump where at the first frame Ag occupies UHUC and at the next frame, one minute later, the Ag atom disappears from UHUC and only weak bright contrast around the corner hole is visible. The Ag atom has two possibilities to jump to the corner hole, from FHUC or from UHUC. It is not possible with 100% certainty, within the used scanning rates, to ascertain from which HUC the single atom has jumped to the corner hole. The estimated values for the frequency prefactor and activation energies for the jump to the corner hole from HUCs are $1.4 \times 10^{+8}$ s$^{-1}$ and 0.828 eV respectively[19] (see also supplementary table 1 and supplementary table 2). According to Sobotík the diffusion process of the Ag atoms on Si surface is not influenced by the tip and tunneling conditions of −2.0 V and 400 nA[12]. In our experiments we used even less influential tunneling conditions i.e., lower current (max 200 nA) and lower voltage (max. −1.8 V).

The reverse process of jumping outside CH is much slower and once the Ag atom jumps into the corner hole it stays there for a long time. Surprisingly no jumps were recorded even during the longest scans of several days i.e., 4 days and 7 h for 17 Ag atoms. Even during the experiment 16 new Ag atoms occupied CHs and the count of all occupied CH by Ag atoms at the end of the experiment changed to 33 and no escape of any new Ag from the CH was registered.

The diffusion is determined by the diffusion pathways and the value of the barrier height together with the frequency pre-factor

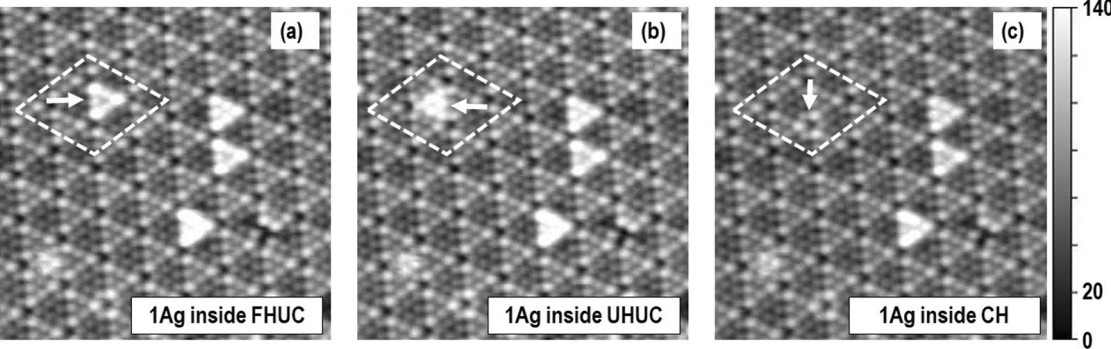

**Fig. 2 Intercell cell hopping and jumping to a corner hole.** Three consecutive filled states STM images (**a**), (**b**), and (**c**) separated by one minute obtained at RT (tunneling conditions: −1.8 V, 177 nA, size: 20×20 nm$^2$). At (**a**) the diffusing Ag atom is denoted by the arrow. The area where the diffusion takes place is denoted by the diamond shape outline. In (**a**) diffusing atom is in FHUC and in (**b**) the same Ag atom is visible in UHUC. In (**c**) the diffusing Ag atom is in CH. In the images we can see one more corner hole that is occupied by the Ag atom and during this process it stays unchanged. Original files in (**a**), (**b**), and (**c**) are provided in Supplementary Data 2: (**a**) m326_ori.par, m326_ori.tf0, m326_ori.tb0. (**b**) m328_ori.par, m328_ori.tf0, m328_ori.tb0. (**c**) m333_ori.par, m333_ori.tf0, m333_ori.tb0.

and adsorption energy can explain low jumping rate[20]. Tip influence on the escape process from the CH could not be excluded based on the lack of escape jumps. The STM tip has the effect on the scanned adatoms through direct interaction, the tunneling current and the high electric field. To further verify the tip influence on the Ag atoms inside CH we performed the experiment with tip retracted from the scanned area. At the beginning of the experiment, we have observed 116 Ag atoms inside CHs on the area 100×100 nm$^2$ at RT (supplementary figure 4a). The tip during the experiment was retracted by 1 μm. After 1 h the tip contact was re-established and the same area of the surface was scanned again and checked if any of the Ag atoms inside CHs that were present earlier inside CH escaped. We found that none out of 116 Ag atoms escaped (see supplementary figure 4b). The experiment clearly shows that the tip influence on the escape rate is negligible.

The difference in the rates of the process of jumping in and outside the corner hole can produce an uneven ratio of atoms with the majority that are inside CH. After scanning the surface for a long time i.e., 4 days and 7 h at the same spot, we saw that the occupancy of HUCs changed and as a result the corner holes count with the Ag atoms inside it. Counting only Ag that did not bound with another Ag atoms in HUC, out of 234 Ag atoms in HUCs, twenty atoms jumped inside CH and none out of nineteen Ag in CHs from the start jumped out (supplementary figure 5). The count of Ag in HUCs changed to 214 Ag atoms. Long time scanning in a UHV chamber was possible without residual gas contamination of the scanning place because of the shading of the STM tip[21].

At a higher temperature of about 150 °C, it was possible to observe the jump of the Ag atom out of the corner hole. The half-lifetime of Ag atom in a corner hole is in the range of minutes (movie 1 suppl. material). Figure 3 shows the diffusion of one Ag atom from one corner hole to the other corner hole through HUCs. In the image, we also can see more corner holes that are occupied with single Ag atoms but during this process they stay unchanged. Analysis of the STM images revealed that the long-time double occupancy of the corner hole with two or more Ag atoms was not observed. Further to this, we conclude that the number Ag atoms adsorbed on the CHs is only one because of the following reasons: (i) the Si(111)-7×7 surface was prepared clean, (ii) on the clean Si(111)7×7 surface, no bright features like on the Ag/Si(111)-7×7 system were visible at all, (iii) the quantity of Ag atoms used during the experiments was also very little, (iv) during the experiments, we did not observe the brightness of two

different kinds i.e., in the STM image we did not observe any other brighter sites other than those with one Ag inside the CH, which if present would suggest the presence of more than one Ag atom, (v) the brightness of the CHs throughout the area under investigation was consistent and the measured height difference distributions follow a normal distribution, which support the claim that the number Ag atoms in these CHs were the same. Finally, a careful comparison of the clean and Ag adsorbed Si(111)-7×7 surface showed increase of the brightness around CH, which is related to the jump made by 1 Ag atom to the CH. As we will present later, our DFT calculations also confirms these experimental observations i.e., increased brightness around the CH with 1 Ag atom inside. At this point we note that it needs further experimental verification and theoretical analysis if more than one atom can occupy the corner hole at the same time as a stable unit.

**Theoretical calculations of Ag atom on the Si(111)-7×7.** To further understand the interaction of Ag adatoms on the Si(111)-7×7 surface we performed quantum chemical calculations based on density functional theory (DFT). Since the Si(111)-5×5 and the Si(111)-7×7 unit cells are similar (Fig. 4(a-b)), to reduce the computational cost, the DFT calculations are first performed on the Si(111)-5×5 surface. Based on the results from Ag/Si(111)-5×5 systems, we repeated the calculation for the Si(111)-7×7 surface only for the most stable site.

Since the experiments suggest that Ag atom is mostly seen at the HUCs and CH we considered the adsorption of Ag adatom on four different sites, which include (i) on the hollow site of the CH, which is a 12-membered Si-ring and (ii) on the hollow site of 8, 6 and 5-membered Si-ring for the comparative purpose (See Fig. 4(a–b) and Supplementary Data 1). Our calculations show that the adsorption energy of Ag on CH is -2.855 eV and on the 8, 6, and 5-membered ring sites it is −1.821 eV, −2.543 eV and −2.235 eV respectively. Clearly, the Ag adatom is most stable on the CH sites agreeing with our experimental findings. In this site, the Ag atom is exactly over the Si atom in the center of the CH from the lower layer of the Si slab, with Si-Ag interatomic distance of 2.391 Å. It is worth noting that, other than on the 5-membered Si-ring in all the other structures the Ag remained in the hollow site (Fig. 4(c–f)). On the 5-membered Si-ring site, the Ag adatom moved to the bridged site (Fig. 4(d)). The most stable structure i.e., Ag on the CH is ~0.3 eV lower in energy than the HUC site. The Ag atom in the CH relaxed to the position directly

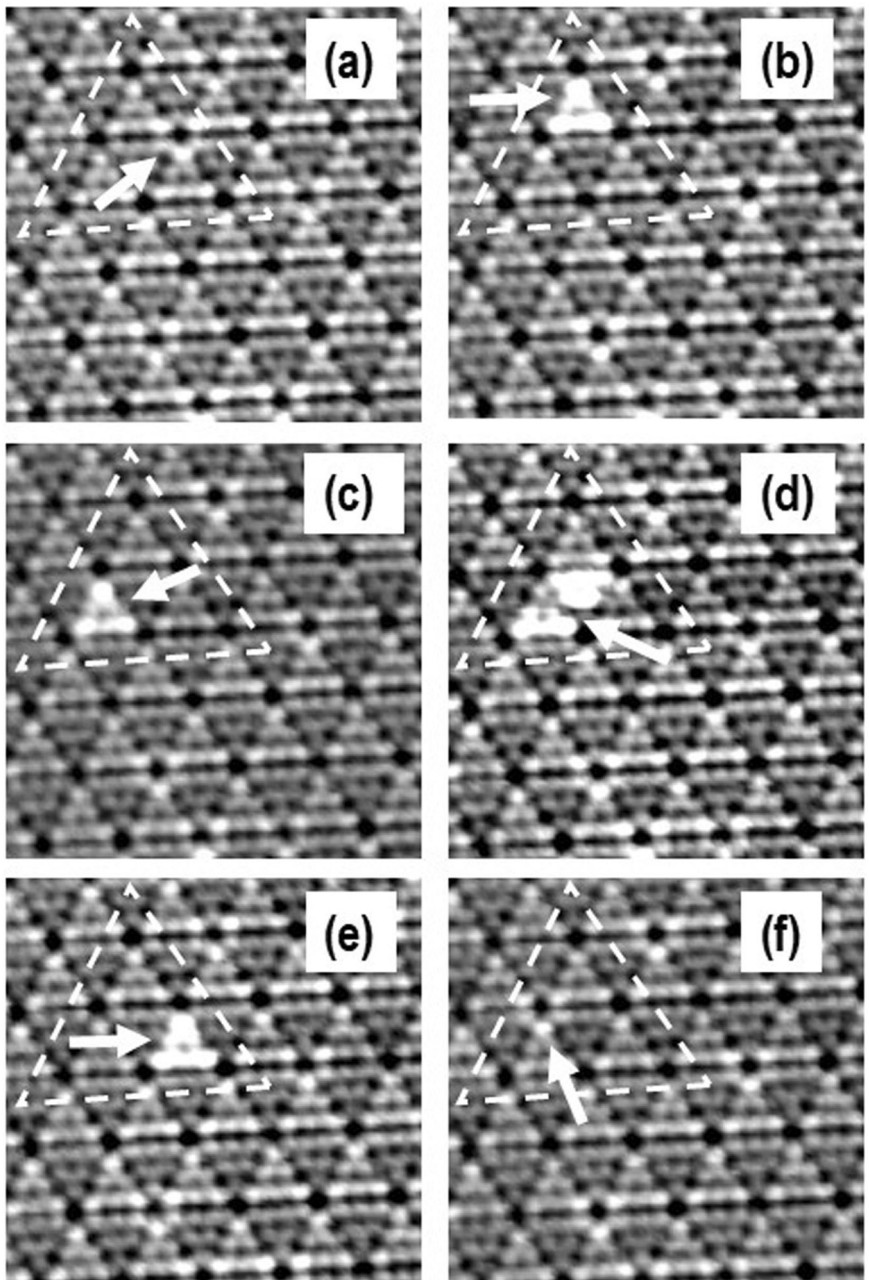

**Fig. 3 Diffusion at 150 Celsius degrees.** The set of six filled states STM images (**a–f**) separated by about 45 s of the same surface area (14×14 nm$^2$, −1.0 V, 166 nA) showing diffusing one Ag atom from one corner hole to the other corner hole through HUCs at about 150 Celsius degrees. The area of interests where diffusion of single Ag atom takes place is outlined by the triangular shape and diffusing atom is denoted by the arrow. In (**a**) Ag atom is in the CH. In (**b**) the Ag atom is in FHUC and in (**c**) the Ag atom is in FHUC. (**d**) Ag is in transition between FHUC and UHUC. In (**e**) the Ag atom is in FHUC. In (**f**) the Ag atom is in CH. In all the images we can see more CHs that are occupied with Ag atoms but during this process they stay unchanged. Original files in (**a–f**) are provided in Supplementary Data 2: (**a**) m112_ori.par, m112_ori.tf0, m112_ori.tb0. (**b**) m113_ori.par, m113_ori.tf0, m113_ori.tb0. (**c**) m114_ori.par, m114_ori.tf0, m114_ori.tb0. (**d**) m115_ori.par, m115_ori.tf0, m115_ori.tb0. (**e**) m116_ori.par, m116_ori.tf0, m116_ori.tb0. (**f**) m120_ori.par, m120_ori.tf0, m120_ori.tb0.

above the silicon atom at the bottom center of the corner hole. Based on these results, we repeated the calculation of Ag adatom on the 12-membered Si-ring for the Si(111)-7×7 unit cell. The adsorption energy for which was -2.799 eV, which is very close to the most stable Ag/Si(111)-5×5 system. The Si–Ag interatomic distance in Ag/Si(111)-7×7 system was found to be 2.392 Å, which is also very close to the values reported above.

In the next step of our theoretical investigation, we focused on understanding the charge transfer phenomenon due to the

adsorption of Ag on the Si(111)-5×5 and the Si(111)-7×7 surface. The partial density of states of the Ag atom on the Si(111)-5×5 and the Si(111)-7×7 surfaces are summarized in Fig. 5, which showed that the majority of the up and down spin states are below the Fermi energy ($E_F$). We believe it is due to the electron gained by the Ag adatom[22–25]. We also found some up and down-spin states above the Fermi energy in Fig. 5(a and b), which may be because of the anti-binding orbitals arising due to the chemical bonding of the Ag atom with the Si-atom below it.

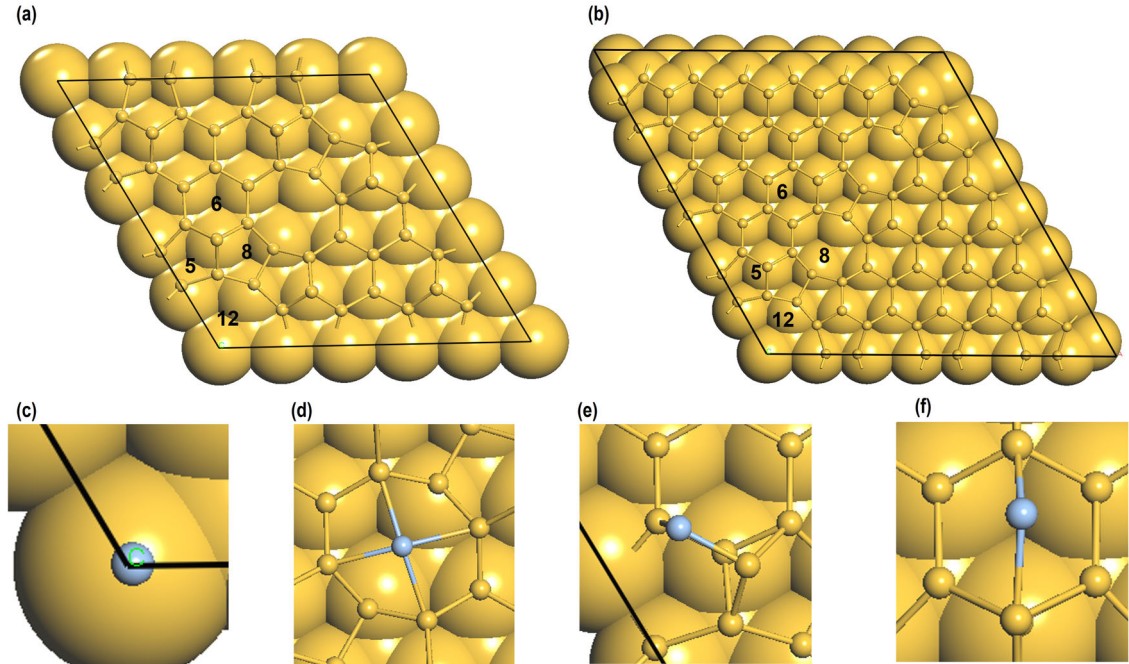

**Fig. 4 Adsorption of Ag atom in different sites.** (**a**) Si(111)-5×5 and (**b**) Si(111)-7×7 unit cell showing the 12, 8, 6, and 5-membered Si rings on the surface. The optimized structures of Ag adatom (represented by light-blue balls) on (**c**) 12, (**d**) 8, (**e**) 6 and (**f**) 5-membered Si rings on the Si(111)-5×5 surface. For clarity the top two-layers of Si atoms (in yellow) are shown as ball and stick, and the lower layers of atoms are shown in the CPK form.

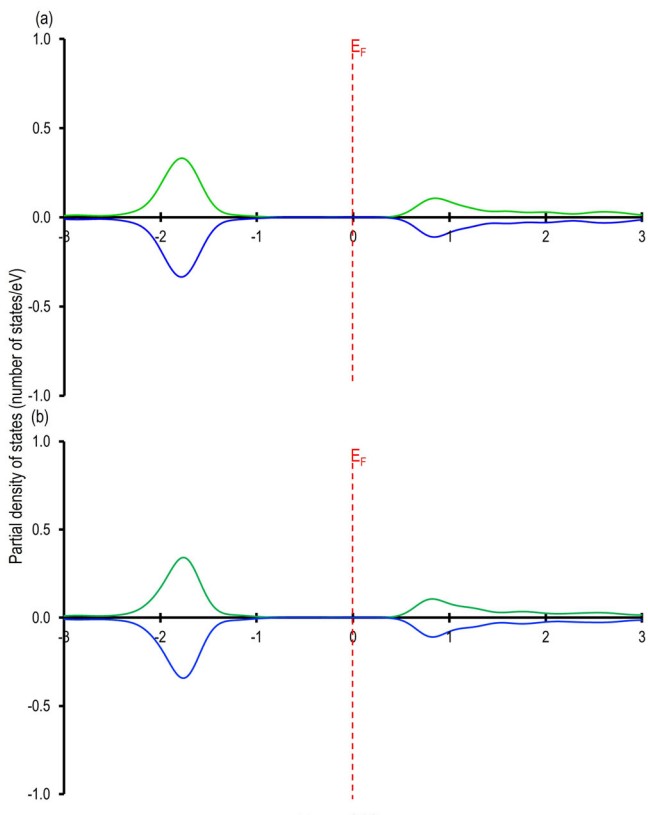

**Fig. 5 Partial density of states of Ag-adatom.** Partial density of states for Ag-adatom on the 12 membered Si-ring of the (**a**) Si(111)-5×5 and (**b**) Si(111)-7×7 unit cell. The Fermi energy ($E_F$) is represented by the dotted line passing through 0 eV and the green and blue line represents the up and down spins of the Ag $s$-orbital signatures respectively. Raw data in (**a**) and in (**b**) is provided in in Supplementary Data 2: (**a**) and (**b**) in DOS_7x7_5x5_AgSi111.xlsx.

To further confirm these findings, we investigated the Bader charges of the Ag-adatom on the CH of the Si(111)-5×5 and the Si(111)-7×7 surfaces, which are -0.111 e and -0.235 e respectively. This agrees well with our analyses on the PDOS and confirms charge transfer from the Si(111)-5×5 and the Si(111)-7×7 surfaces to the Ag-adatom. As shown in Fig. 6, we also performed an analysis on the DFT-derived charge density difference for Ag/Si(111)5×5 and Ag/Si(111)7×7 systems, which showed electron gain on the Ag adatom and electron loss in the Si-atoms just below it. We believe the accumulation of charge on top Ag adatom induces charge redistribution around the CH, which may result in brighter STM images around it. In order to confirm that there is a bond between the Ag and the Si-atom we generated a cluster model from the Ag/Si(111)-7×7 system (Fig. 7 (a)). In this model, the Ag atom, the Si-atom just below it ($Si_{Ag}$) and the first and the second shell of another nine Si-atoms bonded to the $Si_{Ag}$-atom, were considered (Fig. 7(b)). The dangling bonds were saturated with H atoms. To avoid any change in the geometry with respect to the Ag/Si(111)-7×7 system, only the H-atoms were relaxed. Our calculations revealed a Mayer bond order value of 0.915 (~1), which confirmed single bond formation between the Ag and the Si-atoms.

Finally, as shown in Fig. 8, we simulated the STM images of the pristine Si(111)-7×7 and the Ag/Si(111)-7×7 systems. The simulated image showed that on the pristine surface all the surface atoms are equally bright however, on the Ag/Si(111)-7×7 surface the atoms around the corner holes are relatively brighter than the rest of the system, which is similar to the experimental observation i.e. the STM images, where the adatoms around the corner hole with Ag residing inside it are brighter (see Figs. 1–3) than the equivalent site without the Ag atoms. To further understand the reason behind the brightness of the surface Si-atoms around the CHs we analyzed the Bader charges, which reveal that before and after Ag-atom adsorption the average charge on uppermost Si-atoms are -0.04 e and -0.06 e, which show that increase in brightness is related to subtle electron redistribution around the CH on the surface. We believe, this

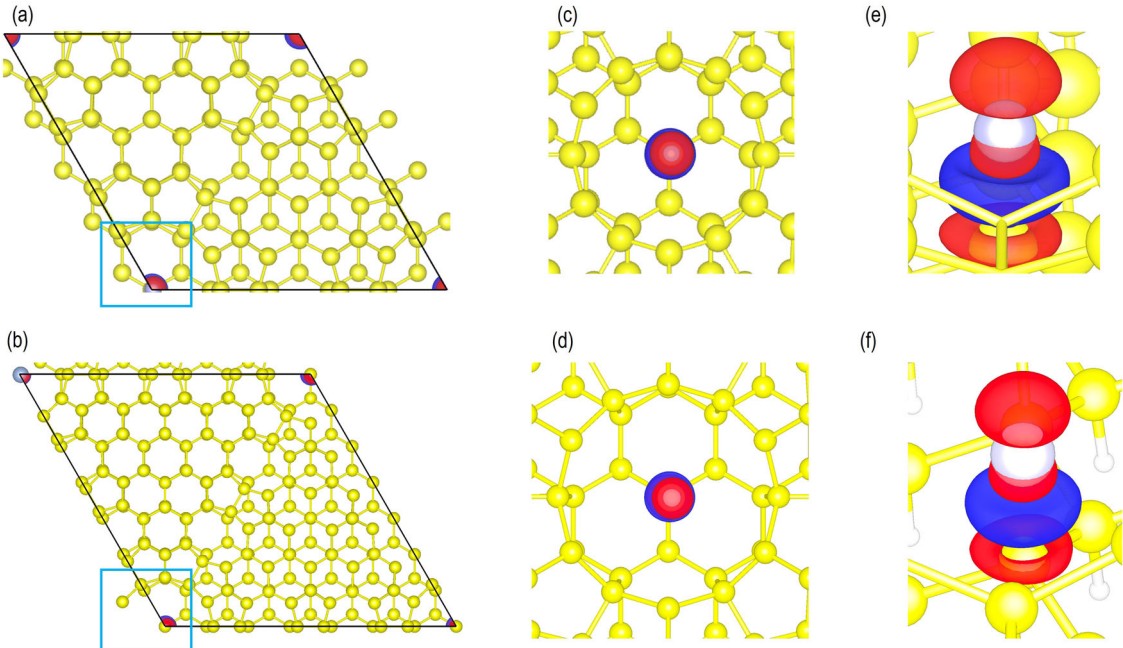

**Fig. 6 Charge density difference of the Ag-adatom in the CH.** The isosurface lobes (0.003 e/Å³) of charge density difference of the Ag adatom adsorbed on (**a**) the Si(111)-5×5 and (**b**) the Si(111)-7×7 unit cell. The blue square shows the CH with the Ag adatom. The top view of the charge density difference on the CH of (**c**) Si(111)-5×5 and (**d**) Si(111)-7×7 unit cells. The side views of the enlarged charge density difference in the CH of (**e**) Si(111)-5×5 and (**f**) Si(111)-7×7 unit cells. Red and blue lobe represents charge gain and charge depletion respectively. The yellow and light-blue balls represent Si and Ag atoms respectively.

finding provides direct theoretical evidence on the experimental observation of a single Ag atom being adsorbed in the corner holes of the Ag/Si(111)-7×7 system.

## Discussion

Based on the above experimental and theoretical results one could design novel protocols by adjusting deposition rate, amount, and heating to force the maximum number of single atoms into CHs in a relatively short time with desired surface density. Already images (a) and (f) in Fig. 3 at 150 °C show that this small area of the surface is only populated with atoms in corner holes. There are also some Ag clusters in HUCs that trap atoms. Thus, corner holes and Ag clusters in HUCs are the main trapping units of single Ag atoms at 150 °C (supplementary figure 6).

The Si(111) surface is very unique and to the best of our knowledge it seems to be the only choice among all pristine semiconducting crystals whose surface reconstructs in such a way that it readily creates relatively deep and small surface pockets that are also a part of the basic unit cell for long time atom entrapment at RT. This interesting phenomenon has remained unnoticed for a long time. The reason being it is experimentally difficult to observe as the adsorption site at the CH is below the surface at the third atomic layer. Further to this, most of the experiments are performed naturally in RT conditions. Therefore, the process of an atomic jump into the corner hole at this temperature is not so easy to observe because of its low rate. Even if brighter features are visible around the CH at the beginning, they cannot be easily assigned to the presence of an atom without observing the jump process. Therefore, such features were probably considered as some sort of contamination that should be discarded in the analysis.

Comparing the RT result with 150 °C clearly shows the dependency of the escape process on the temperature with activation energy. Cooling the surface down below RT will further

extend the lifetime of the species inside CH and might provide an opportunity of building long-time stable real devices that are based on a single atom at a corner hole.

The corner holes that are like pockets might be a good way for trapping other elements and while choosing a particular element, one should always take into consideration that it does not diffuse into the bulk of Si crystal easily[26]. There are already good candidates that might behave like Ag i.e., Cu and Au[27] as the size of the corner hole is significantly large. We, therefore, theoretically explored the interaction of Li, Na, Cu, Au, F, and I on the Si(111)-5×5 surface to test if other atoms would also behave in a similar fashion like Ag. The calculated adsorption energies show that except for Cu and Li all the other atoms are most stable in the CHs (See Table 1). In the case of the Cu/Si(111)-5×5 and the Li/Si(111)-5×5 systems, the stability on the CH and the six-membered holes are comparable. Further to this, we found that in the case of Cu-adatom on the 8-membered ring there is surface restructuring, and it diffuses to the bulk. The Si–Si bond distance in the vicinity of the diffused Cu-adatom increases slightly to accommodate it in the bulk. Experimental data support the fact that Cu diffuses interstitially and stays in the interstitial site in thermal equilibrium at the diffusion temperature[28].

We note that the Si(111) surface also can reconstruct itself to a 5×5 structure with the same corner hole entities as on the 7×7 surface. The 5×5 surface gives a readily about the twice denser population of corner holes than 7×7 and might provide an alternative to the 7×7 surface for controlled placement of atoms inside the corner hole however, the 5×5 surface is harder to obtain and is still not well explored for surface diffusion like the 7×7 surface. In order to form 5×5 on the pristine Si(111) it needs some additional preparation effort[29]. Further to this, it is worth noting that the dangling bond at the bottom of the corner hole might serve as a ready-made anchor and mediator of electronic interaction with silicon crystal. The atom movement in the corner hole can be comparable to the atomic movement of an encapsulated atom

**(a)**

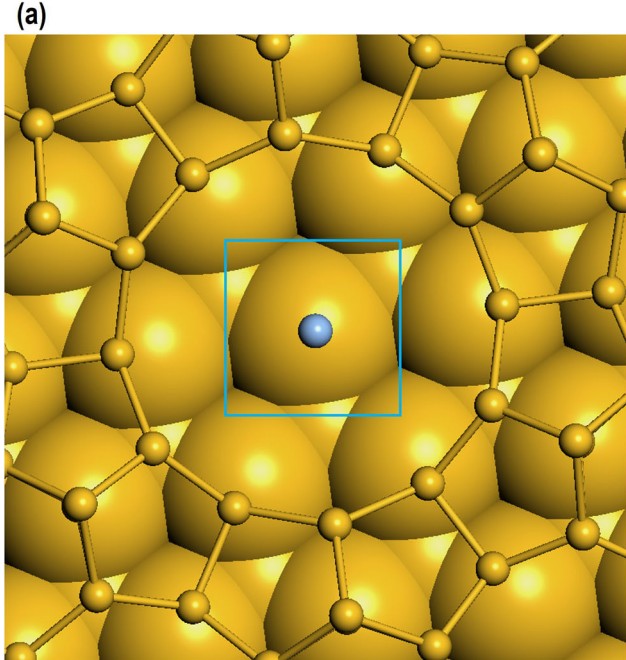

**(b)**

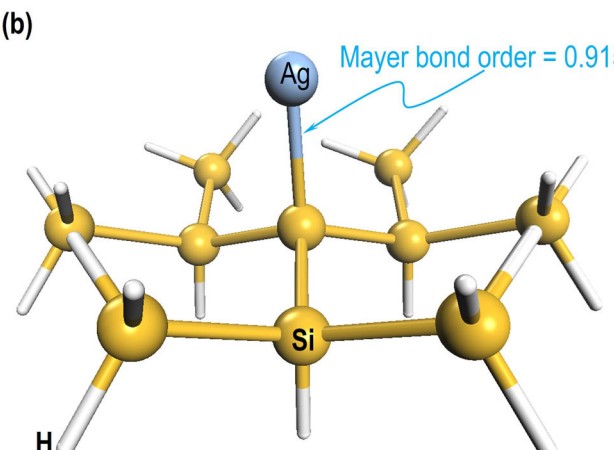

Mayer bond order = 0.915

Ag

Si

H

**Fig. 7 Bond order between Ag-adatom and Si-atom in the CH.** (**a**) The fully relaxed Ag/Si(111)-7×7 system. The blue rectangle shows the position of Ag atom in the CH and (**b**) AgSi$_{10}$H$_{21}$ cluster model obtained from (**a**) for the calculations on Mayer bond order. The Si and Ag-atoms are represented by yellow and light blue balls and for clarity the H-atoms in (**b**) are represented by white sticks.

inside C60[30]. An experimental and theoretical investigation is needed to understand atomic dynamics inside CH.

Atoms that will have non-zero nuclear magnetic moment in the corner hole seem to be quite attractive species for making qubits based on the spin of the nucleus[31]. The spin current that can be arranged to flow through the atom can exert a torque on the nuclear spin through hyperfine interaction of the electron and nucleus. Other interactions like quadrupole nuclear moments or atomic dipole moments that could form in the hole can be used. Also, the vibrational modes of an atom inside a CH might be exploited for building such a device. Experimental validation of such propositions can entirely be performed using scanning tunneling microscope[31].

The fact that Ag atoms stay for a very long time in a corner hole at RT is favorable for building a device. Moreover, double occupancy of the corner hole by two Ag atoms so far has not been

**(a)**

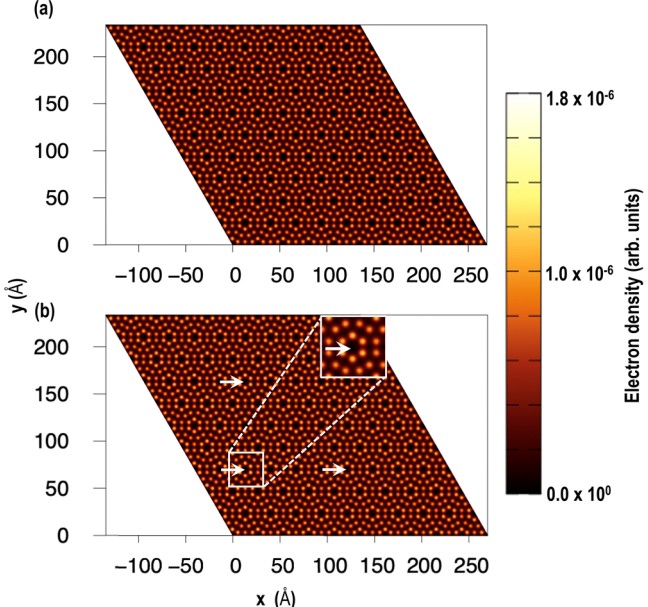

**Fig. 8 Simulated STM image of Si(111)7×7 surface with and without Ag-adatom.** The simulated STM images of (**a**) pristine Si(111)-7×7 and (**b**) Si(111)-7×7 with Ag atoms in the corner hole. The Ag atoms in the CHs are indicated using arrows and can be recognized by the surrounding six top adatoms that belong to the six adjacent unit cells and those adatoms look bright (inset figure in (**b**)).

**Table 1 The adsorption energies of Cu, Ag, Au, Li, Na, F, and I on the Si(111)-5×5 surface.**

| X/Si(111)-5x5 | Holes | E$_{ad}$ (eV) |
|---|---|---|
| Cu | 10 | −3.240 |
|  | 8 | −3.060 |
|  | 6 | −3.246 |
|  | 5 | −2.455 |
| Ag | 10 | −2.855 |
|  | 8 | −1.821 |
|  | 6 | −2.543 |
|  | 5 | −2.235 |
| Au | 10 | −3.694 |
|  | 8 | −2.565 |
|  | 6 | −3.405 |
|  | 5 | −3.296 |
| Li | 10 | −2.562 |
|  | 8 | −2.119 |
|  | 6 | −2.564 |
|  | 5 | −2.076 |
| Na | 10 | −2.307 |
|  | 8 | −1.810 |
|  | 6 | −2.095 |
|  | 5 | −1.688 |
| F | 10 | −6.215 |
|  | 8 | −3.900 |
|  | 6 | −4.450 |
|  | 5 | −3.552 |
| I | 10 | −3.553 |
|  | 8 | −1.974 |
|  | 6 | −3.250 |
|  | 5 | −1.743 |

detected, which gives hopes for the production of regular surface arrays of separated single atomic units. Even possible long-time entrapment and close separation of two single atoms in nearby corner holes might be interesting in the view of entanglement and

vacuum and surface-mediated atomic interactions[32]. The possibility of atomic manipulation further makes it more attractive.

The building of the device based on the corner hole should address the issue of the chemical activity of Si(111)-7×7 by some way of protecting it from undesirable conditions. For instance, the 7×7 structure does not survive exposure to the atmosphere[33] therefore, protection techniques such as building a cover for example using interesting materials like graphene or some other materials might be useful[34]. Interestingly adsorbed molecules, that act as a top cage that entraps the atom for life inside CH even at high temperatures that have not yet been explored for such purpose, might prove useful for building stable air-tight useful devices[35].

## Methods

**Experiment**. Experiments are performed in UHV. The UHV system is composed of two separate chambers i.e., the preparation chamber and the STM chamber. The base pressure in the chambers is lower than $1.5 \times 10^{-8}$ Pa and lower than $2 \times 10^{-8}$ Pa during Ag deposition. The evaporation of Ag is performed either by e-beam heating or thermal evaporation using hot filament. The sample substrates are cut from the commercial n-type Si(111) wafers (phosphorus-doped, $1 \div 10$ Ohm cm). Clean and well-ordered Si(111)-7×7 surfaces are prepared by heating the sample substrate by direct current up to 1490 K (5 times for about 5 s) in the sample preparation chamber. In the final step the sample is quickly cool down from 1490 K to 1200 K and from 1200 K till room temperature at a rate of about 1 K/s. Ag atoms are deposited at a rate lower than $5 \times 10^{-3}$ monolayer (ML) per second. 1 ML is defined as the density of Si atoms on a Si(111) surface, i.e., 7.81 atoms/cm$^2$. Direct counting of the Ag atoms is used to calibrate the deposition rates. After the deposition the tunneling contact is established in about 10 to 20 min. The STM operates in constant current mode. The drift correction in the STM enables us to obtain images of the same area. We use W tips that are prepared by the electro-chemical etching.

**Computational details**. We have performed quantum mechanical calculations[36–38] within the framework of density functional theory (DFT) as implemented in the Vienna Ab initio Simulation Package. We employed the projector augmented wave (PAW) method[39] and the exchange and correlation functional within the generalized gradient approximation by Perdew, Burke, and Ernzerhof (PBE)[40]. The wave functions are expanded in a plane wave basis with an energy cutoff of 500 eV, which gives bulk energies converging to $10^{-5}$ eV. The six atomic layered Si(111) 5×5 and the Si(111)-7×7 models were generated from a Si unit cell with a theoretically determined lattice parameter of 5.452 Å, which is close to the experimental value of 5.431 Å. In the Si(111) 5×5 model, there are 100 Si atoms and 25 H atoms to terminate the bottom Si layer. Similarly, in the Si(111)-7×7 model there are 200 Si atoms, and 49 H atoms. The Si atoms on the bottom two layers are fixed and all the other atoms are fully relaxed until the forces are less than 0.01 eV/Å. To avoid slab-slab interaction a vacuum of ~15 Å was used along the z-axis. Furthermore, the spurious dipole moments due to the adsorption of the Ag atoms on the exposed surfaces of the above models were taken into account by using the methods of Neugebauer et al.[41,42]. The adsorption energies ($E_{ad}$) are calculated using the equation:

$$E_{ad} = E_{Total\ system} - E_{pristine} - E_{Ag\ atom}, \quad (1)$$

where, $E_{Total\ system}$, $E_{pristine}$ and $E_{Ag\ atom}$ are the total energy of the Ag/Si(111)5×5 or the Si(111)-7×7 system, total energy of the pristine Si(111)5×5 or the Si(111)-7×7 surfaceand Ag atom respectively. The charges on various atoms were obtained using the Bader charge analysis as implemented by Henkelman and co-workers[43]. The charge density difference, $\rho_{diff}$ was calculated using

$$\rho_{diff} = \rho_{Total\ system} - \rho_{pristine} - \rho_{Ag\ atom}, \quad (2)$$

where, $\rho_{Total\ system}$, $\rho_{pristine}$ and $\rho_{Ag\ atom}$ are the charge of the Ag/Si(111)5×5 or the Si(111)-7×7 system, total energy of the pristine Si(111)5×5 or the Si(111)-7×7 surface and Ag atom respectively. In this study we have also employed Grimme's dispersion correction (D3) as dispersive effects might be significant for these systems[44,45].

To explore the bond order of the adsorbed Ag atom on the Si(111)-7×7 surface, the spin polarized DFT calculations on the AgSi$_{10}$H$_{21}$ cluster model was performed using the ORCA code[46]. Since the AgSi$_{10}$H$_{21}$ cluster was extracted from the fully relaxed Ag/Si(111)-7×7 system, the dangling bonds were saturated with H-atoms and constrained optimization was performed i.e., only the H-atoms are relaxed. For this calculation, the triple-zeta valence with polarization version of the Ahlrichs def2 basis set (def2-TZVP) with effective core potential for Ag-atoms, PBE exchange and correlation functional and Grimme's D3 dispersion was employed[40,44–47].

Finally, we used the Quantum Espresso package[48] to simulate the STM images of the pristine and Ag-adsorbed Si(111)-7×7 systems, which is based on the Tersoff-Hamann approximation that assumes that the STM current is proportional to the local density of states[49]. For these simulations, we used the relaxed geometries from the VASP calculations, and the STM maps were calculated at 1.5 eV and at a constant height mode. The simulated STM images were visualized using the Critic 2 package[50,51].

## Data availability

The data supporting the finding of the study are included in the article and Supplementary Information files, additional data are available from the corresponding authors upon request.

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

## Acknowledgements

This work was partially supported by Grants-in-Aid (No. JP17340091) for Scientific Research from the Ministry of Education, Culture, Sports, Science and Technology (MEXT) and by the 21st Century Center of Excellence (COE) program "Exploring New Science by Bridging Particle-Matter Hierarchy" from MEXT. In addition, this study was partially supported by JSPS KAKENHI Grants (No. JP15K05119 and JP19K03681). AC acknowledges the use of the Cirrus UK National Tier-2 HPC Service at EPCC (funded by University of Edinburgh and EPSRC (EP/P020267/1), ARCHER and ARCHER-2 (via Materials Chemistry Consortium, and EPSRC (EP/L000202)), and Athena at HPC Midlands + (via RAP Call 2019 EP/P020232/1). J.O acknowledges MAX IV Laboratory for the use of the STM on Bloch Beamline. Research conducted at MAX IV, a Swedish national user facility, is supported by the Swedish Research council under contract 2018-07152, the Swedish Governmental Agency for Innovation Systems under contract 2018-04969, and Formas under contract 2019-02496. J. R. O expresses gratitude to C. M. Polley for help to set up the STM microscope at Bloch Beamline.

## Author contributions

Conceptualization: J.O., A.C. Methodology: J.O., A.C., S.S. Investigation, experiment: J.O. Investigation, calculations: A.C. Funding acquisition: J.O., S.S., A.C. Project administration: S.S., A.C. Supervision: S.S., A.C. Writing—original draft: J.O., A.C. Writing—review & editing: J.O., S.S., A.C.

## Competing interests

The authors declare no competing interests.
