## [Peer Review File · Nature Communications]

REVIEWER COMMENTS

Reviewer #1 (Remarks to the Author):

The paper demonstrates that Ag atoms can be trapped in the corner holes of the Si(111)-7x7 surface and remain there for a long period of time (days) at room temperature. The authors suggest that the reason Ag atoms are stable in corner holes is due to the electronegativity of the bond, based on theoretical calculations.

The extended lifetime of Ag atoms trapped in the corner holes of the Si(111)-7x7 reconstruction could potentially be a useful means of fabricating a periodical array of Ag atoms on a silicon surface. As the authors outline, this result could potentially provide a route to nanoscale device manufacturing.

The authors seem to suggest that this is the first observation of metal atoms being trapped in corner holes, and this claim may well be true. However, it has been demonstrated in the past that the potential energy landscape of the Si(111)-7x7 surface is very strongly site-specific (D Lock et al 2015 J. Phys.: Condens. Matter 27 054003), and as such the observation that Ag atoms have an affinity for adsorption at a specific site of the surface is not unexpected.

A few questions that I would like the authors to answer:

1. The experiments seem to rely on the continuous scanning of the Si surface decorated with Ag atoms. What is the mechanism of diffusion into the corner holes? Is it purely thermal, or assisted thermal? What is the role of the physical presence of the tip? What about the tunnelling electrons/holes?
2. Figure 5 shows a calculation of the partial density of states of Ag atoms on Si(111)-5x5. In both subfigures, there appear to be energy states below -2 V. Are all experimental results obtained at a scanning bias of -1.8 V? If so, would there be any electron-induced dynamics due to the presence of low-energy states in the Ag atoms? What is the nature of the Ag bond on the Si surface? Are the Ag atoms chemisorbed or physisorbed? Is there a difference between different sites?
3. How did you choose -1.8 V as the scanning voltage? What happens at lower electron energies, where the electrons are less likely to affect the dynamics of Ag atoms?
4. Did you do any time-lapse measurements to check the stability of the Ag atoms in corner holes without the presence of the STM tip and tunnelling current?

5. Are the activation energy and prefactor on line 167 experimentally or theoretically estimated? Can you estimate them from your experiments?

6. You show a couple of diffusion pathways in Figures 2 & 3. Can you estimate the rate of these transitions across the unit cells? Are transitions from the faulted or the unfaulted half of the unit cell more likely or of the same likelihood?

7. Line 181: can you clarify what “some shading of the STM tip” means?

8. Cooling the sample down is suggested as a possible way of increasing the lifetime of the trapped Ag atoms in the corner holes. However, it is well-known that cooling down the Si(111)-7x7 reconstruction dramatically effects the pinning of the Fermi level and hence the energy landscape of the Si surface. To compensate for this temperature effect, highly doped samples are required. How would sample doping affect the dynamics of Ag on the Si surface and the stability of Ag trapping in corner holes?

9. Nuclear decay of atomic isotopes has been suggested as a possible application. Has this ever been done before and is it even possible?

10. Can the authors present some evidence as to why they think that there is only 1 Ag atom involved in the dynamics that they observe?

11. The suggestion of stabilising the surface reconstruction outside vacuum by covering it with a sheet of graphene seems a bit contradictory, considering that the presence of graphene on top of the created structures would completely change the electronic properties of both Ag and the Si(111)-7x7 surface.

12. Can the authors provide some experimental scanning tunnelling spectroscopy results to support their theoretical results in Figure 5?

- Additional comments:

Please state the scanning parameters for all STM images in Figures 1-4.

Please correct spelling and grammar across paper. Caption of Figure 1 is wrong.

Lines 167-171; 211-214; 308-311 confusing.

Reviewer #2 (Remarks to the Author):

Referee report

Re: NCOMMS-21-15675

Atoms love holes, by Jacek R. Osiecki, Shozo Suto, Arunabhiram Chutia

The work presented in this manuscript reports the entrapment of a silver atom in a hole on a Si(111)-7x7 surface. The findings are supported by both experimental and theoretical investigations and support the capability of atomic manipulation at room temperature. According to the conclusions, engineering holes that can trap the manipulated atoms is expected to realize single atom based devices. The authors present the details of the adsorption and trapping including diffusion of the trapped atom. The atomic entrapment on a specific site is well presented in this work. Although the work is clearly presented and is original, its high interest is not clear, as well as how it advances the field. The authors should first address the comments below, before the manuscript can be accepted for publication in Nature Communications.

Main Concern:

The experimental results rely only on STM images, while these are not really correlated with the theoretical results shown in the manuscript. Taking into account the target journal, a more in detail analysis would be expected (see points 2 and 5 below for example). This could include more evidence on the STM images (see point 1. below) and the calculations, as well as a direct comparison of the two. At this point, the simulations do not provide any insightful evidence towards the experimental results. Overall, more rich and insightful results would be expected for Nat. Commun.

Other Comments:

1. The authors claim that the higher brightness of the holes is related with the presence of an Ag atom. This observation is only based on their STM images and the partial density of states from the simulations. How certain can this claim be? Do the authors have other arguments for this? How can they be sure that the condition of the tip is not affecting the images, thus the claims? Since brighter

features are visible around the corner hole (CH) and could be considered as some sort of contamination, could this also be the case at the CH?

2. Can the authors provide with more information on diffusion, i.e. diffusion constants?

3. Do the authors have evidence of the stability of the surface and the tip during the 4 days of scanning?

4. The authors discuss that "all atomic elements from Mendeleev's table are viable options", but this cannot be the case, as also not all surfaces or Si surfaces provide the pockets for the adsorption of these atoms. A more detailed discussion on the relevant elements and surfaces would be more insightful.

5. In the paper the atomic dynamics inside the CH are not addressed. In view of a more complete study, this would be a very important point. Could the authors provide some results on this?

6. The discussion on an outlook and relevance of this investigation is quite vague. How would a "quantum single atomic device" function and what would be its purpose? How would the protection of a Si surface be related with the current investigation in order to build "stable useful devices"? How do these applications really connect to the presented work?

Minor comments:

(a). The manuscript should be checked throughout for grammar and typesetting errors.

(b). The term "unfavorable condition" for the tip should be clarified.

(c). Have the authors checked that the 5x5 reconstructed Si surface in the simulations leads qualitatively to the same results as the 7x7 used in the experiments? Have they also benchmarked the simulation parameters, i.e. exchange-correlation functional?

Reviewer #3 (Remarks to the Author):

1 Key results: Please summarise what you consider to be the outstanding features of the work.

This paper reports on the deposition of Ag atoms onto Si(111)-7x7, the stability at different sites, and the study at elevated temperature of the surface using STM.

This is nice STM. The surface is well prepared, the images are clear, with long range order that is defect free. The presented results belie the complexity and hard work required to produce repeated atomic resolution STM scans over the timescales studied, and with adsorbed atoms. There is no question that these are nice data and well-performed experiments.

2 Originality and significance: If the conclusions are not original, please provide relevant references. On a more subjective note, do you feel that the results presented are of immediate interest to many people in your own discipline, and/or to people from several disciplines?

There is already work looking at adsorbates on the corner hole sites of Si(111)-7x7, particularly with aromatic molecules, which are stably bound at RT and can only be removed by atomic manipulation using the STM tip. There has to be a discussion of prior work looking at adsorbate bonding to the corner hole site, since this is the entire focus of the paper. At the moment the authors report just one study [3] which is not sufficient. They do go on later to report elements adsorbed on the surface and say that the corner hole site was not studied. However the work of Zhang ref 21 (PRL 94, 176104 (2005)) looks at Ag on Si(111)-7x7 using STM – exactly the same as the authors. This work is cited, but to say that Cu and Au behave similarly – seemingly not acknowledging that Zhang studied Cu, Au, and importantly also Ag. Zhang doesn't neglect to study the corner hole site, they specifically say they don't believe bonding is occurring there. The authors should at the very least compare their results to this.

The authors cite ref 5 to say Ag is spatially localised at LT. However a quick read indicates this paper is H on Si(100)-2x1?

In the opening, the authors report that phosphine gas can be stabilised on the hydrogen terminated Si(111)-7x7 surface, but that this requires UHV conditions. The experiments presented here are in UHV too. Is direct deposition of Ag (their method) more suitable to non UHV conditions? I understand it is a simpler method, but don't know why they are arguing that UHV is the problem, when they also use UHV. I can see what they are saying, because I understand creating a templated surface is more complicated; my point is that this would be unclear to a non-specialist.

The main issue with significance is in the title though. The authors have not studied all atoms and cannot use a title which implies that all atoms "love holes". When publishing work on site-specific adsorbates on surfaces, it is customary to include some statistical analysis to back up any claim of site preference. We need to know how many sites were observed, and whether any observed difference is statistically significant. I understand their claim on "love" to be referring to the half life once bound to the corner hole site. I appreciate a snappy title, but wonder here if even with Ag added, it doesn't adequately explain what they've done.

3 Data & methodology: Please comment on the validity of the approach, quality of the data and quality of presentation. Please note that we expect our reviewers to review all data, including any extended data and supplementary information. Is the reporting of data and methodology sufficiently detailed and transparent to enable reproducing the results?

Ag atoms on corner hole sites make the surrounding six adatoms appear brighter. "The bottom of the corner hole with Ag is only about a few pico meters higher than the empty one, depending on the state of the tip. In case of adatoms surrounding CH they are "higher" by about 5 to 20 pm (fig. 2 suppl. mat.)." In this figure it's a little hard to interpret the line profile, because it looks like the indicated line on the right section goes further than the corresponding line profile does in the left? It might be helpful to mark the atom positions that are being studied. However, the main problem is how inexact this measurement is "higher by about 5 to 20 pm", and how this is reported in the main text. Since this is the main mechanism that is being studied, and the paper depends on knowing that there is definitely still an Ag atom present, (and that this might disagree with existing work by Zhang) we cannot rely on subjective interpretation of what looks brighter. Particularly because in supp Fig. 2 the leftmost atom in the linescan is actually brighter than one of the supposedly brighter atoms in the raised adatoms because of the Ag. With the existing data, there is nothing to stop the authors analysing all the line scans over the studied sites and quantitatively assessing the distribution of 'bright' adatoms compared to the distribution of heights of the background ones. Thus an independent test of significance could be created which quantitatively decides that a ring is bright enough to indicate the presence of a Ag atom. I think from the image, it probably is brighter, but it's so close to call that a non-subjective method would be preferred. The authors acknowledge that the tip state can remove the contrast difference completely, which further shows that they must prove what 'bright' is beyond doubt.

The authors say that atoms adsorbing to the corner hole post deposition are “roughly about 10%”. Please give the actual numbers involved. It is normal to say you counted x adsorbed atom sites, and y were in corner holes, z were not. Then give the exact percentage. Similarly when you discuss inter-cell jumping, it would be normal to give the exact number of observations, over a specific timescale, and categorise all the jumps detected quantitatively. Later some numbers are given, but these should come earlier and it isn’t clear if the statements made earlier are based on the same observations? Later you say you scanned the surface for “about 4 days and 7 hours”. If it was 4 days and 7 hours, this isn’t “about”?

How did you calculate the estimated half-life of Ag in the corner hole site at RT?

When you say “We have performed theoretical calculations” it would be normal to give the method used in the text. Later in the main text you say DFT, but why not when first discussed?

Cooling to 0 K would pin all adsorbates. I don’t see how that’s useful. Surely the benefit of your observation is that there is a long lifetime at RT?

The impact of the work at the end starts to go a bit far. Do the authors know that the CHs don’t interact with free electrons? The final sentence of this paragraph doesn’t make sense: “The silicon surface also might also be intentionally doped to such an extent to modulate the conduction and as a result electronic components of the interaction.” And when the authors start talking about adding a graphene top cage to maintain the 7x7 reconstruction in atmosphere, this has moved completely beyond the work presented in the paper. There are interesting results here, but I don’t think all of the claimed significance is consistent with the work presented.

The authors state that atomic species do not diffuse into the bulk of Si. Just from memory, atomic O does insert beneath the Si(111)-7x7 surface when adsorbed. From memory it is the work of Mayne and Djurdin in Paris.

“The fact of this atomic behavior somehow has been unnoticed and neglected for a long time and an explanation lies in the difficult experimental observation and that adsorption site at CH is below the surface below the third atomic layer”. This is a bold claim and I don’t think substantiated.

4 Suggested improvements: Please list additional experiments or data that could help strengthening the work in a revision.

Although there are only a few mistakes in the English, the construction is non-standard and would need editing prior to publication. There is also repetition, sometimes in consecutive sentences. The abstract is particularly challenging, and so is addressed in more detail in the next section.

5 Clarity and context: Is the abstract clear, accessible? Are abstract, introduction and conclusions appropriate?

The opening line of the abstract is not sufficiently convincing on why nanotechnology depends on single atom manipulation and stabilization.

Some of the results in the abstract are first reported vaguely (we report long time confinement) and then later reported with more detail also within the abstract (trapped...range of days at RT; minutes at 150 C). I would suggest removing the 'previews' and just reporting detail. I would suggest rather than approximate time scales, the actual measured values are reported, removing ambiguity (about 150 C; range of days; several minutes; a quite closed space). There is detail in the last paragraph of the introduction and I would expect to see this in the abstract.

From the abstract I don't understand the position the Ag atom is fixed at on the 7x7 reconstruction from the line "inside at the bottom center". I don't understand why C60 is mentioned? I am aware that some of these points are explained further in the manuscript itself, but the authors should remember that readers get to the abstract first, which must be self-contained. There are standard conventions for naming sites on this reconstruction, and the authors go on to say that the Ag atom is trapped at a corner hole. Is this the same site implied from the preceding sentence, or this is a different aspect of the study?

After already demonstrating that there is temperature dependence to the entrapment by reporting results, the abstract then states this, afterwards? It is stated that the thermal activation process is reduced through cooling the surface, but have these experiments be done at LT? It isn't clear what is established fact being reported from the literature, and what are new claims based on the results of this work.

I am not sure the four sentences at the end of the abstract which claim the usefulness of this work are supported by the results presented before, at least in the current form.

On the introduction, the last section gives a full summary of results. Is this the correct place for this?

6 References: Does this manuscript reference previous literature appropriately? If not, what references should be included or excluded?

This is discussed before, in terms of setting out existing work and comparing these results to it.

7 Please indicate any particular part of the manuscript, data, or analyses that you feel is outside the scope of your expertise, or that you were unable to assess fully.

I am not sure about the appropriateness of only modelling the 5x5 unit cell. Hopefully someone else will comment. Although it used to be challenging computationally, certainly I have read recent papers that model the full unit cell of 7x7. There are other modelled systems that could be compared to their results.

8 Please address any other specific question asked by the editor via email.

Response to the Reviewers

REVIEWER COMMENTS

Reviewer #1 (Remarks to the Author):

The paper demonstrates that Ag atoms can be trapped in the corner holes of the Si(111)-7x7 surface and remain there for a long period of time (days) at room temperature. The authors suggest that the reason Ag atoms are stable in corner holes is due to the electronegativity of the bond, based on theoretical calculations.

We thank the reviewer for his insightful comment and apologize for the misunderstanding. We believe that the stability of the Ag atom on the corner holes is because of the electron transfer from Si to the Ag atom. To clarify this point further, we have now added a discussion on the bond order between the Ag adatom and the Si-atom of the Si(111)-7x7 surface. Our calculation shows that the bond order is 0.901 (~1), which confirmed single bond formation between the Ag and the Si-atoms.

The extended lifetime of Ag atoms trapped in the corner holes of the Si(111)-7x7 reconstruction could potentially be a useful means of fabricating a periodical array of Ag atoms on a silicon surface. As the authors outline, this result could potentially provide a route to nanoscale device manufacturing.

We thank the reviewer for her/his nice comment and understanding and we have now mentioned this in the main text.

The authors seem to suggest that this is the first observation of metal atoms being trapped in corner holes, and this claim may well be true. However, it has been demonstrated in the past that the potential energy landscape of the Si(111)-7x7 surface is very strongly site-specific (D Lock et al 2015 J. Phys.: Condens. Matter 27 054003), and as such the observation that Ag atoms have an affinity for adsorption at a specific site of the surface is not unexpected.

We thank the reviewer for her/his comment and we have looked into the paper, which discusses the adsorption of aromatic molecules such as benzene, toluene and chlorobenzene on the Si(111)-7x7 surface. However, our study is on the entrapment of Ag adatom on the corner hole of the Si(111)-7x7 surface for a long time (even for days). We believe our study is different, but we have decided to include the significant work done by Lock et al in the main text. Further to this, we would like to highlight that using DFT calculations we have shown clearly that adsorption of Ag adatom is site specific and as mentioned in the previous comment we have provided theoretical reasons behind this, which has not been reported earlier. Additional DFT based calculations have been also provided for several other elements to support our study.

A few questions that I would like the authors to answer:
1. The experiments seem to rely on the continuous scanning of the Si surface decorated with Ag atoms. What is the mechanism of diffusion into the corner holes? Is it purely thermal, or assisted thermal? What is the role of the physical presence of the tip? What about the tunnelling electrons/holes?

The manuscript is devoted to the adsorption of the Ag atom inside the corner hole and their long stability there. It is true that the electrons that come from the tip and tip itself might cause some excitations like e.g. vibrations, electronic excitations etc. However, here the STM worked in the image collection mode on a relatively large area like 100 nm x 100 nm, with low current

Response to the Reviewers

density in the range of less than 200 nA. The tip is spending very little time at every site while scanning. Moreover, long scanning for several days did not cause the escape of the Ag atoms from the corner hole on a relatively large population at all.

Moreover, in publication by P Sobotík, P Kocán, I Ošťádal, "Direct observation of Ag intra-cell hopping on the Si(111)-(7×7) surface", Surface Science, Volume 537, Issues 1–3, 2003, Pages, L442-L446, [https://doi.org/10.1016/S0039-6028\(03\)00608-3](https://doi.org/10.1016/S0039-6028(03)00608-3), there was no tip influence observed with even "worse" tunneling conditions like -2.0V and 0.4 nA. The reference to this paper has been provided and in the main text we have mentioned it clearly as per the suggestion of the reviewer.

Moreover, Zhang et al. in [21] did not register during experiments activation of the diffusion of the Ag atoms inside the HUC by STM tip. There, inside the HUC the diffusion barriers are lowest and still the Ag atom was observed at one place with -2.0 V (filled states, higher voltage than in our experiment) when cooled down. This fact is also convincing that imaging of the Ag atoms without any appreciable influence is possible while scanning with relatively low current density.

Based on the above references we believe that the diffusion to the corner hole in our experiments is thermally activated with negligible tip influence.

These details, based on the comment by the reviewer, have been now included in the main text.

2. Figure 5 shows a calculation of the partial density of states of Ag atoms on Si(111)-5x5. In both subfigures, there appear to be energy states below -2 V. Are all experimental results obtained at a scanning bias of -1.8 V? If so, would there be any electron-induced dynamics due to the presence of low-energy states in the Ag atoms? What is the nature of the Ag bond on the Si surface? Are the Ag atoms chemisorbed or physisorbed? Is there a difference between different sites?

We agree with the reviewer but most of the experiments were performed with -1.8V and lower till -1.0 V. This is true that electrons might tunnel to some states that are below -2V and induce some dynamics. However, since the highest tunneling voltage is -1.8 V, this was not observed i.e. any observed influence on the escape rate from the corner hole at RT. The influence on the dynamics inside CH is possible but we have not studied it here as it is out of the scope of this article, but it is briefly mentioned as a future experimental suggestion.

According to the publication by Zhang et. al. [21] the Ag intercell diffusion inside the HUC basin, where diffusion barriers are very small, the imaging with the STM did not cause the movement of the Ag atom. Please also read the response in 1 above.

Further to this we have now included some more theoretical calculations and their discussion on the bond order in the main text. Please refer to Figure 7 and the main text. For the convenience of the reviewer, we present it below as well.

"In order to confirm that there is a bond between the Ag and the Si-atom we generated a cluster model from the Ag/Si(111)-7x7 system (Figure 7 (a)). In this model, the Ag atom, the Si-atom (Si_{Ag}) just below it and the first and the second shell of another nine Si-atoms bonded to the Si_{Ag} -atom, were considered (Figure 7(b)). The dangling bonds were saturated with H-atom. To avoid any change in the geometry with respect to the Ag/Si(111)-7x7 system, only the

Response to the Reviewers

H-atoms were relaxed. Our calculations revealed a Mayer bond order value of 0.901 (~1), which confirmed single bond formation between the Ag and the Si-atoms.”

Figure 7. (a) The fully relaxed Ag/Si(111)-7x7 surface and (b) AgSi₁₀H₂₁ cluster model obtained from (a) for the calculations on Mayer bond order.

3. How did you choose -1.8 V as the scanning voltage? What happens at lower electron energies, where the electrons are less likely to affect the dynamics of Ag atoms?

The tunneling voltage -1.8V was an appropriate voltage to see the bright features around the CHs in this particular tip. However, at this voltage the probability to influence the dynamics i.e. escape from the CH was not observed. At the lower voltage, the visibility of Ag inside CH is less apparent. For a full explanation on this issue please read our response to the comments in 1 and 2.

4. Did you do any time-lapse measurements to check the stability of the Ag atoms in corner holes without the presence of the STM tip and tunnelling current?

As it is shown in responses to 1, 2, and 3 which are related to this question, the tip influence is negligible. As we can see that Ag atoms are stable during scanning for a very long scan and literally no escape was observed. Therefore, no time-lapse measurements were made for this study to exclude influence of the tip. It would be reasonable and logical to think that if some jumps were recorded then it would have been necessary to check the influence via time-lapse measurements. This explanation has been included in the main text.

5. Are the activation energy and prefactor on line 167 experimentally or theoretically estimated? Can you estimate them from your experiments?

We thank the reviewer for this important question. The activation energy and prefactor is cited from the reference (19) and it was experimentally determined. The results were not published in any journal previously and we felt it is a good opportunity to reference it here to make it more public. Therefore, in addition to citing this work as: [19] Jacek Osiecki, Atomistic diffusion and

Response to the Reviewers

clustering of Ag atoms on a well defined Si(111)7x7 surface studied by STM, *PhD Thesis*, (2007), we have also added more information in the supplementary part as “Supplementary, Excerpt from thesis”.

6. You show a couple of diffusion pathways in Figures 2 & 3. Can you estimate the rate of these transitions across the unit cells? Are transitions from the faulted or the unfaulted half of the unit cell more likely or of the same likelihood?

It is already known that the transition from the unfaulted HUC to the faulted HUC is more likely to happen and we provide references. From the above, we cannot assume anything about the preference of the transition to the corner hole. Since the Ag atom is spending more time inside the FHUC, the transition from the FHUC is more likely to be observed within a given set.

To conclude, it is not determined within this work from which HUC the jump to the corner hole is more likely to happen. Since the rates of the jumps are relatively low it is reasonable to assume that the jump occurred from the HUC in which Ag was just observed in the previous scan. What we want to say here is that one cannot determine with 100% probability from which HUC the jumps occurred. However, we plan to study this phenomenon in greater detail in the next study. We believe the lack of such information here does not change the main arguments of the article as it is mainly about the stability of the Ag atoms in the CHs.

7. Line 181: can you clarify what “some shading of the STM tip” means?

It is known that shading of the tip is happening because of the size of the tip. In UHV there is residual gas that can find a way to the surface. If it is blocked by the tip, there is a shading effect of the tip. One can see the stability of the surface caused by the shading in the supplementary figure 3. Two sets of images are separated in time by 4 days and 7 hours. The scanning was continuous.

Based on the reviewer's comment now we have added one reference in which the importance of the shading of the tip was mentioned and deposition of material was performed at an angle to avoid tip shading.

[21] Ošťádal Ivan, Kocán Pavel, Sobotík Pavel, Pudl Jan, Direct Observation of Long-Range Assisted Formation of Ag Clusters on Si(111)-7x7, *Phys. Rev. Lett.* 95, 146101, (2005)

8. Cooling the sample down is suggested as a possible way of increasing the lifetime of the trapped Ag atoms in the corner holes. However, it is well-known that cooling down the Si(111)-7x7 reconstruction dramatically effects the pinning of the Fermi level and hence the energy landscape of the Si surface. To compensate for this temperature effect, highly doped samples are required. How would sample doping affect the dynamics of Ag on the Si surface and the stability of Ag trapping in corner holes?

We thank the reviewer for the very insightful comment. We believe the above argument about the Fermi level of the electrons is indeed worth investigating. But if the atomic escape is thermally activated, cooling will surely lower the escape rate. Therefore, the full picture of energy transfer to the atom is worth exploring but it is out of the scope of this article.

9. Nuclear decay of atomic isotopes has been suggested as a possible application. Has this ever been done before and is it even possible?

Response to the Reviewers

We thank the reviewer for the question and after careful thinking we have decided to remove this sentence. We, however, note that depositing atomic isotopes was just a proposition for future application and might be interesting to perform.

10. Can the authors present some evidence as to why they think that there is only 1 Ag atom involved in the dynamics that they observe?

As reported in Figure 2, we showed that only one Ag atoms jumps from the HUC to the CH. These experiments were repeated many times and we have not observed more than one Ag atom in the process. We would also like to note that every experiment started from a clean surface where we did not observe such features due any contamination, which is very strong evidence that only 1 Ag atom is involved.

11. The suggestion of stabilising the surface reconstruction outside vacuum by covering it with a sheet of graphene seems a bit contradictory, considering that the presence of graphene on top of the created structures would completely change the electronic properties of both Ag and the Si(111)-7x7 surface.

We thank the reviewer for the insightful comment but this has been put in the paper just as a future proposition only. The detailed electronic properties of both Ag with graphene and Si(111)-7x7 have not yet been explored, but it might be of interest to the readers. However, complying with the reviewers comment we have slightly modified the sentence to clarify it as shown below:

“Building of the device based on the corner hole should address the issue of chemical activity of Si(111)-7x7 by some way of protecting it from undesirable conditions. For instance, the 7x7 structure does not survive exposure to the atmosphere²⁷ therefore, protection techniques such as building a cover for example using interesting materials like graphene or some other materials²⁸ might be useful.”

12. Can the authors provide some experimental scanning tunnelling spectroscopy results to support their theoretical results in Figure 5?

We do not have spectroscopy results at this moment however, we plan to perform such experiments at low temperature. More theoretical and experimental studies are being carried out at this moment.

- Additional comments:

Please state the scanning parameters for all STM images in Figures 1-4.

Figure 1. a) -1.8, 177nA , b) -2.0V, 153nA

Figure 2. -1.8V, 177nA

Figure 3. -1.0 V, 166 nA, sample at 150 C deg.

Please correct spelling and grammar across paper. Caption of Figure 1 is wrong.

We thank the reviewer for the comment and now it has been modified.

Response to the Reviewers

Lines 167-171;

The lines 167- 171 has been modified as follows:

“The diffusion is determined by the diffusion pathways and the value of the barrier height together with the pre factor can explain the lowest jumping rate ²¹. According Sobotík et. al. the diffusion process is not influenced by the tip and tunneling conditions of -2.0 V and 400 nA ¹². “

211-214; Corrected.

308-311 confusing.

Corrected text to: “References”

Response to the Reviewers

Reviewer #2 (Remarks to the Author):

Referee report

Re: NCOMMS-21-15675

Atoms love holes, by Jacek R. Osiecki, Shozo Suto, Arunabhiram Chutia

The work presented in this manuscript reports the entrapment of a silver atom in a hole on a Si(111)-7x7 surface. The findings are supported by both experimental and theoretical investigations and support the capability of atomic manipulation at room temperature. According to the conclusions, engineering holes that can trap the manipulated atoms is expected to realize single atom based devices. The authors present the details of the adsorption and trapping including diffusion of the trapped atom. The atomic entrapment on a specific site is well presented in this work. Although the work is clearly presented and is original, its high interest is not clear, as well as how it advances the field. The authors should first address the comments below, before the manuscript can be accepted for publication in Nature Communications.

We thank the reviewer for his positive comment and we address all her/his comments point by point below.

Main Concern:

The experimental results rely only on STM images, while these are not really correlated with the theoretical results shown in the manuscript. Taking into account the target journal, a more in detail analysis would be expected (see points 2 and 5 below for example). This could include more evidence on the STM images (see point 1. below) and the calculations, as well as a direct comparison of the two. At this point, the simulations do not provide any insightful evidence towards the experimental results. Overall, more rich and insightful results would be expected for Nat. Commun.

We thank the reviewer for comments and we will clarify the referee's doubts in the following responses.

Other Comments:

1. The authors claim that the higher brightness of the holes is related with the presence of an Ag atom. This observation is only based on their STM images and the partial density of states from the simulations. How certain can this claim be? Do the authors have other arguments for this? How can they be sure that the condition of the tip is not affecting the images, thus the claims? Since brighter features are visible around the corner hole (CH) and could be considered as some sort of contamination, could this is also be the case at the CH?

We provided detailed images in Figures 1, 2 and 3 in the main text and the hopping of the Ag to the CH. The surface is prepared as clean as it can be and no such features as the corner hole with Ag was observed before depositing Ag atoms in countless experiments we performed, which proves that the brightness of the holes is related with the presence of an Ag atom.

Response to the Reviewers

Even if the state of tip is influencing the images, the data that we provide shows that there is Ag inside CH without a doubt just by comparing images before and after the jump. The process was observed many times and the images representing Ag inside CH always look the same i.e. brighter features around the CH. On the other hand, the DFT based theoretical calculations clearly showed that the Ag atom was most stable in the CH, agreeing well with the experimental observation. We have now added more theoretical calculations to show that the stability of the Ag atom on the CH is because of the formation of chemical bond. Please refer to page 8 and 9 in the main text.

2. Can the authors provide with more information on diffusion, i.e. diffusion constants?

Since we are not discussing the long-distance diffusion of Ag but more on the change of the presence of Ag inside the HUC and the CH as a single process (a jump) such diffusion constants have not been reported. We believe the discussion on the diffusion constants are out of the scope of the paper. However, we have provided the jumping rate to the CH from the HUC. Please refer to page 6 in the main text.

3. Do the authors have evidence of the stability of the surface and the tip during the 4 days of scanning?

Please refer to the supplementary figure 3. As evidence we provided two sets of images, which were separated in time by 4 days and 7 hours. The scanning was continuous and no contamination is visible in the images separated by such a long scan. If the surface was left alone without the STM tip inside the UHV chamber considerable contamination would be visible.

4. The authors discuss that "all atomic elements from Mendeleev's table are viable options", but this cannot be the case, as also not all surfaces or Si surfaces provide the pockets for the adsorption of these atoms. A more detailed discussion on the relevant elements and surfaces would be more insightful.

We thank the reviewer for her/his suggestion and based on her/his suggestion we have now modified the main text where we have omitted the above statement. We, however, note that the size of the atomic pocket (CH) is large enough to contain some other atoms. For example, we have now included another table with more results from DFT based quantum chemical calculations to show that in addition to Ag atoms, the CHs could be also favorable for elements such as Li, Na, Cu, Au, F and I.

5. In the paper the atomic dynamics inside the CH are not addressed. In view of a more complete study, this would be a very important point. Could the authors provide some results on this?

We thank the reviewer for his excellent comment; however, we feel that this could be a good starting point for the next study that addresses the atomic motion inside CH. Certainly the analysis of atomic movement inside CH with quantum chemical molecular dynamic simulations would give a complete picture. At this stage we believe such calculations and supporting experiments are out of the scope of this article.

6. The discussion on the outlook and relevance of this investigation is quite vague. How would a "quantum single atomic device" function and what would be its purpose? How would the protection of a Si surface be related with the current investigation in order to build "stable useful

Response to the Reviewers

devices"? How do these applications really connect to the presented work?

We apologize for the vagueness of the statement and now based on her/his comments we have modified the main text. We note that the content of this study is mainly focused around the atomic entrapment on the Si(111)-7x7 surface and we believe that it is extremely important on its own. We are not presenting any devices, but we gave some probable future propositions for further experimental/theoretical studies.

Minor

comments:

(a). The manuscript should be checked throughout for grammar and typesetting errors.

We once again apologize for the grammatical and typesetting errors. We have now carefully revised the manuscript as much as we can and we are open to further improving the language.

(b). The term "unfavorable condition" for the tip should be clarified.

The state of the tip is unknown but at the present state-of-the-art of equipment it is normal to operate the STM in such a way. The state of the tip can be changed with voltage pulsing till one can see the proper quality and the contrast. To conclude, the operator of the STM does not know the state of the tip and what kind of atom terminates the tip. The discussion about the influence of the state of the tip on the image is rather out of the scope of this article. It can be addressed in the future if experimental setups will allow it.

We want to stress that images of the Ag atoms with some contrast are not hard to get. It is rather easy and requires adjustment of the voltage and possibly some pulsing of the tip.

(c). Have the authors checked that the 5x5 reconstructed Si surface in the simulations leads qualitatively to the same results as the 7x7 used in the experiments? Have they also benchmarked the simulation parameters, i.e. exchange-correlation functional?

We thank the reviewer for her/his comments and based on which we have now added results from Si(111)-7x7 surface as well. Please refer to the main text with all the modifications. For her/his convenience we present a brief abstract from the main text below:

“To further understand the interaction of Ag adatoms on the Si(111)-7x7 surface we performed quantum chemical calculations based on density functional theory. Since the Si(111)-5x5 and the Si(111)-7x7 unit cells are similar (Figure 4 (a – b)), to reduce the computational cost, the DFT calculations are first performed on the Si(111)-5x5 surface. Based on the results from Ag/Si(111)-5x5 systems, we repeated the calculation for the Si(111)-7x7 surface for the most stable site.”

So far as the benchmarking of the calculations are concerned, as reported in the computational detail we have determined the theoretical lattice parameters for the Si unit cell based on which the Si(111)-5x5 and Si(111)-7x7 surfaces were constructed. The theoretically determined

Response to the Reviewers

lattice parameter is 5.452 Å, which is close to the experimental value of 5.431 Å using GGA/PBE exchange and correlation functional. Additionally, we have used Grimme's dispersion corrections, as the van der Waals correction may be important for such studies. The PBE exchange and correlation functionals are used as they are well-established functional and are widely used for such studies. Further to this, the theoretical calculations were performed in order to understand the nature of interaction between Ag and the Si(111)-5x5 and the Si(111)-7x7 surfaces. Therefore, no separate studies on the relevance of other exchange and correlation functionals were carried out as it is beyond the scope of this study.

Response to the Reviewers

Reviewer #3 (Remarks to the Author):

1 Key results: Please summarise what you consider to be the outstanding features of the work.

This paper reports on the deposition of Ag atoms onto Si(111)-7x7, the stability at different sites, and the study at elevated temperature of the surface using STM.

This is nice STM. The surface is well prepared, the images are clear, with long range order that is defect free. The presented results belie the complexity and hard work required to produce repeated atomic resolution STM scans over the timescales studied, and with adsorbed atoms. There is no question that these are nice data and well-performed experiments.

2 Originality and significance: If the conclusions are not original, please provide relevant references. On a more subjective note, do you feel that the results presented are of immediate interest to many people in your own discipline, and/or to people from several disciplines?

There is already work looking at adsorbates on the corner hole sites of Si(111)-7x7, particularly with aromatic molecules, which are stably bound at RT and can only be removed by atomic manipulation using the STM tip. There has to be a discussion of prior work looking at adsorbate bonding to the corner hole site, since this is the entire focus of the paper. At the moment the authors report just one study [3] which is not sufficient. They do go on later to report elements adsorbed on the surface and say that the corner hole site was not studied. However the work of Zhang ref 21 (PRL 94, 176104 (2005)) looks at Ag on Si(111)-7x7 using STM – exactly the same as the authors. This work is cited, but to say that Cu and Au behave similarly – seemingly not acknowledging that Zhang studied Cu, Au, and importantly also Ag. Zhang doesn't neglect to study the corner hole site, they specifically say they don't believe bonding is occurring there. The authors should at the very least compare their results to this.

We thank the reviewer for the comment and as suggested by the referee we have now summarized the most important finding of our study in the main text.

Our work is devoted to atomic entrapment inside the CH not aromatic molecules. Aromatic molecules studied before are too big to go inside the CH and they are adsorbed on the top of the surface, not in the adsorption site inside the CH. This is an entirely different situation to what we are presenting. The publication by Zhang that is given by the referee is devoted to the adsorption of the elements inside the HUC not the CH. Now based on new theoretical calculations we further understand that other atoms like Au, Cu, Li, Na, F and I behave in a similar fashion inside the CH.

The authors cite ref 5 to say Ag is spatially localised at LT. However a quick read indicates this paper is H on Si(100)-2x1?

We thank the reviewer and we apologize for the confusion. Actually, what we meant is that the hydrogenated surface has all dangling bonds saturated with hydrogen⁶ and by selectively removing the hydrogen atoms and by exposing to the specific gas e.g. phosphine gas can result in phosphorus atoms in desirable positions after heat treatment³.

Response to the Reviewers

To support our statement, we merely referred to reference 6, which described this process. To avoid further confusion we have modified the text.

In the opening, the authors report that phosphine gas can be stabilised on the hydrogen terminated Si(111)-7x7 surface, but that this requires UHV conditions. The experiments presented here are in UHV too. Is direct deposition of Ag (their method) more suitable to non UHV conditions? I understand it is a simpler method, but don't know why they are arguing that UHV is the problem, when they also use UHV. I can see what they are saying, because I understand creating a templated surface is more complicated; my point is that this would be unclear to a non-specialist.

We again apologize for the misunderstanding. In the main text we mentioned that adsorption of a single atom or a molecule on the Si surface is possible, but they may require a UHV setup. Based on the reviewers comment we have now modified the sentence for clarity to the non-specialist.

The main issue with significance is in the title though. The authors have not studied all atoms and cannot use a title which implies that all atoms "love holes". When publishing work on site-specific adsorbates on surfaces, it is customary to include some statistical analysis to back up any claim of site preference. We need to know how many sites were observed, and whether any observed difference is statistically significant. I understand their claim on "love" to be referring to the half life once bound to the corner hole site. I appreciate a snappy title, but wonder here if even with Ag added, it doesn't adequately explain what they've done.

We thank the reviewer for his comment. We believe that the title of this article is appropriate but based on her/his suggestion we performed some more theoretical calculations based on DFT, which include Li, Na, Cu, Au, I, and F on different adsorption sites on the Si(111)-5x5 surface. Our calculations displayed similar behavior as with Ag on the Si(111)-5x5 and the Si(111)7x7 surfaces. All the results have been now included in the main text and we believe that we have been now able to justify the title of the paper.

3 Data & methodology: Please comment on the validity of the approach, quality of the data and quality of presentation. Please note that we expect our reviewers to review all data, including any extended data and supplementary information. Is the reporting of data and methodology sufficiently detailed and transparent to enable reproducing the results?

Ag atoms on corner hole sites make the surrounding six adatoms appear brighter. "The bottom of the corner hole with Ag is only about a few pico meters higher than the empty one, depending on the state of the tip. In case of adatoms surrounding CH they are "higher" by about 5 to 20 pm (fig. 2 suppl. mat.)." In this figure it's a little hard to interpret the line profile, because it looks like the indicated line on the right section goes further than the corresponding line profile does in the left? It might be helpful to mark the atom positions that are being studied. However, the main problem is how inexact this measurement is "higher by about 5 to 20 pm", and how this is reported in the main text. Since this is the main mechanism that is being studied, and the paper depends on knowing that there is definitely still an Ag atom present, (and that this might disagree with existing work by Zhang) we cannot rely on subjective interpretation of what looks brighter.

Particularly because in supp Fig. 2 the leftmost atom in the linescan is actually brighter than one of the supposedly brighter atoms in the raised adatoms because of the Ag. With the

Response to the Reviewers

existing data, there is nothing to stop the authors analysing all the line scans over the studied sites and quantitatively assessing the distribution of 'bright' adatoms compared to the distribution of heights of the background ones. Thus an independent test of significance could be created which quantitatively decides that a ring is bright enough to indicate the presence of a Ag atom. I think from the image, it probably is brighter, but it's so close to call that a non-subjective method would be preferred. The authors acknowledge that the tip state can remove the contrast difference completely, which further shows that they must prove what 'bright' is beyond doubt.

This is normal in the STM experiment that the state of the tip might be different every time the new experiment is performed. Such little variations in height and brightness with the presence of the Ag are inevitable. As suggested by the reviewer, we have changed figure 2 in supplementary material to make it clearer to check the relative positions and easier to compare the line profile and the image.

Further to this, we note that detailed images in Figures 1, 2 and 3 in the main text provide evidence of the hopping of the Ag to the CH. The Si(111)-7x7 surface was prepared as clean as it can be and before the deposition of the Ag atoms, no such bright features as the corner hole with Ag was observed. We performed this experiment countless times, which proves that the brightness of the holes is related with the presence of an Ag atom. Please refer to Figure S1 in the supplementary information.

The authors say that atoms adsorbing to the corner hole post deposition are "roughly about 10%". Please give the actual numbers involved. It is normal to say you counted x adsorbed atom sites, and y were in corner holes, z were not. Then give the exact percentage. Similarly when you discuss inter-cell jumping, it would be normal to give the exact number of observations, over a specific timescale, and categorise all the jumps detected quantitatively. Later some numbers are given, but these should come earlier and it isn't clear if the statements made earlier are based on the same observations? Later you say you scanned the surface for "about 4 days and 7 hours". If it was 4 days and 7 hours, this isn't "about"?

We thank the reviewer for her/his insightful comment and based on her/his suggestions we have revised the data and now we have provided the exact count of the atoms and the exact percentage. Moreover the word "about" is now removed. The main text has been modified.

How did you calculate the estimated half-life of Ag in the corner hole site at RT?

We thank the reviewer for the comment and we make the decision not to give an estimation of the half-lifetime. Now we change the main text and provide the information that reflects the experiment directly.

In the abstract it is now:

"We found that a single metal Ag atom can be trapped inside the corner hole even at room temperature for a long time and 17 Ag atoms stayed entrapped for 4 days and 7 hours."

In the main text the places where we present half-lifetime are also changed to provide more information."

When you say "We have performed theoretical calculations" it would be normal to give the

Response to the Reviewers

method used in the text. Later in the main text you say DFT, but why not when first discussed?

We thank the reviewer for his comment. Based on the reviewer's comment we have now mentioned the use of density functional theory based quantum chemical calculations as the theoretical method at the very outset.

Cooling to 0 K would pin all adsorbates. I don't see how that's useful. Surely the benefit of your observation is that there is a long lifetime at RT?

We thank the reviewer for the comment and now the text has been changed as suggested by the reviewer.

The impact of the work at the end starts to go a bit far. Do the authors know that the CHs don't interact with free electrons? The final sentence of this paragraph doesn't make sense: "The silicon surface also might also be intentionally doped to such an extent to modulate the conduction and as a result electronic components of the interaction." And when the authors start talking about adding a graphene top cage to maintain the 7x7 reconstruction in atmosphere, this has moved completely beyond the work presented in the paper. There are interesting results here, but I don't think all of the claimed significance is consistent with the work presented.

We thank the reviewer for the comment and based on her/his suggestion, we have now modified the main text and we have omitted the above sentence to avoid misunderstanding.

The authors state that atomic species do not diffuse into the bulk of Si. Just from memory, atomic O does insert beneath the Si(111)-7x7 surface when adsorbed. From memory it is the work of Mayne and Djurdjin in Paris.

We apologize for the misunderstanding and based on the reviewer's comments we have modified the main text and focus on the main finding of the experiment and the theoretical calculations. Further to this, we performed some DFT based theoretical calculations on Li, Na, Cu, Au, F and I and we have found that in the we found that in the case of Cu-atom on the 8-membered ring there is surface restructuring, and it diffuses to the bulk. The Si-Si bond distance in the vicinity of the diffused Cu-atom increases slightly to accommodate it in the bulk. Experimental data support the fact that Cu diffuses interstitially and stays in the interstitial site in thermal equilibrium at the diffusion temperature²⁵.

"The fact of this atomic behavior somehow has been unnoticed and neglected for a long time and an explanation lies in the difficult experimental observation and that adsorption site at CH is below the surface below the third atomic layer". This is a bold claim and I don't think substantiated.

The 7x7 structure is well known but this fact of atomic entrapment inside of CH is still unclear to the scientific community. In this article we have provided some clues as to why it happens and therefore, we believe it is reasonable to claim that the phenomenon of atomic entrapment in the CH was somehow unnoticed.

4 Suggested improvements: Please list additional experiments or data that could help strengthening the work in a revision.

Response to the Reviewers

Although there are only a few mistakes in the English, the construction is non-standard and would need editing prior to publication. There is also repetition, sometimes in consecutive sentences. The abstract is particularly challenging, and so is addressed in more detail in the next section.

We apologize for the grammatical errors in the paper. We have now carefully revised the manuscript as much as we can and we are open to further improving the language.

5 Clarity and context: Is the abstract clear, accessible? Are abstract, introduction and conclusions appropriate?

The opening line of the abstract is not sufficiently convincing on why nanotechnology depends on single atom manipulation and stabilization.

Thanks to the reviewer for his comment and based on which we have now modified the abstract. We believe that the ultimate ability to assemble devices atom by atom is somewhere at the limits of nanotechnology. Further we believe that advancement in the Nanotechnology world to a large extent does depend on an ability to manipulate material at the atomistic level as the famous scientist R. P. Feynman envisioned in his famous talk [1].

Some of the results in the abstract are first reported vaguely (we report long time confinement) and then later reported with more detail also within the abstract (trapped...range of days at RT; minutes at 150 C). I would suggest removing the 'previews' and just reporting detail. I would suggest rather than approximate time scales, the actual measured values are reported, removing ambiguity (about 150 C; range of days; several minutes; a quite closed space). There is detail in the last paragraph of the introduction and I would expect to see this in the abstract.

We have now modified the abstract as suggested by the reviewer. Many thanks for the comment.

From the abstract I don't understand the position the Ag atom is fixed at on the 7x7 reconstruction from the line "inside at the bottom center". I don't understand why C60 is mentioned? I am aware that some of these points are explained further in the manuscript itself, but the authors should remember that readers get to the abstract first, which must be self-contained. There are standard conventions for naming sites on this reconstruction, and the authors go on to say that the Ag atom is trapped at a corner hole. Is this the same site implied from the preceding sentence, or this is a different aspect of the study?

We have now modified the text as suggested by the reviewer. The C60 is still in the text as we believe it is vital to mention here to have some point of size reference.

After already demonstrating that there is temperature dependence to the entrapment by reporting results, the abstract then states this, afterwards? It is stated that the thermal activation process is reduced through cooling the surface, but have these experiments be done at LT? It isn't clear what is established fact being reported from the literature, and what are new claims based on the results of this work.

Response to the Reviewers

We thank the reviewer for the comment but we believe that in our experiments we showed that the process is thermally activated by providing experimental data at RT and 150 degrees. Lowering the temperature thus will lower hopping frequency and it is not necessary to perform additional experiments at low temperature to further prove the point.

I am not sure the four sentences at the end of the abstract which claim the usefulness of this work are supported by the results presented before, at least in the current form.

The abstract has been now modified. Please refer to the main text.

On the introduction, the last section gives a full summary of results. Is this the correct place for this?

Text is adjusted and the introduction has been updated as suggested by the reviewer.

6 References: Does this manuscript reference previous literature appropriately? If not, what references should be included or excluded?

This is discussed before, in terms of setting out existing work and comparing these results to it.

We believe that it is fixed now and all above corrections will satisfy this point made by the reviewer.

7 Please indicate any particular part of the manuscript, data, or analyses that you feel is outside the scope of your expertise, or that you were unable to assess fully.

I am not sure about the appropriateness of only modelling the 5x5 unit cell. Hopefully someone else will comment. Although it used to be challenging computationally, certainly I have read recent papers that model the full unit cell of 7x7. There are other modelled systems that could be compared to their results. We thank the reviewer for her/his comments based on which we have included some more results and modified the text as follows:

“To further understand the interaction of Ag adatoms on the Si(111)-7x7 surface we performed quantum chemical calculations based on density functional theory. Since the Si(111)-5x5 and the Si(111)-7x7 unit cells are similar (Figure 4 (a – b)), to reduce the computational cost, the DFT calculations are first performed on the Si(111)-5x5 surface. Based on the results from Ag/Si(111)-5x5 systems, we repeated the calculation for the Si(111)-7x7 surface only for the most stable site.”

We further note that our results showed that the adsorption energy for the Ag atom on the CH of Si(111)-7x7 surface was -2.799 eV, which is very close to the most stable Ag/Si(111)-5x5 system. The Si–Ag interatomic distance in Ag/Si(111)-7x7 system was found to be 2.392 Å, which is also very close to the values reported above. We believe this proves the appropriateness of using the 5x5 model. We have added all these results in the main text.

8 Please address any other specific question asked by the editor via email.

N/A

Reviewers' comments:

Reviewer #1 (Remarks to the Author):

The authors have answered most of my original concerns and have improved the paper by adding additional simulation results. However, there are a few points that I would still like to see better addressed by the authors:

1. Effect of the tip: While I accept the arguments provided in the response by the authors that the tunnelling electrons should not affect the dynamics of the adsorbed Ag atoms at the scanning bias, it would be interesting to see whether or not the continuous scanning affects the stability of the Ag atoms in CHs at room temperature. Is it possible that the presence of the tip above the scanning area somehow aids the stability of Ag atoms in CHs, either through an electric field mechanism, or an effect similar to the 'tip shading'? It would be useful to see statistical measurements of the sample surface occupancy with Ag atoms taken at significant time intervals (hours, days) without continuous scanning in order to completely disentangle any effect that the tip may have.
2. Many of the figures in the main text and in the supplementary material are still missing image sizes and scanning parameters.
3. The added excerpt from a thesis in the supplementary material lacks any information about how the data was obtained and how many experiments were performed.
4. As reviewer 3 pointed out, the title is still too general. Even though the added simulations suggest that there might be more atoms that behave in the same way as Ag, it is certainly not the case for O atoms for example. I believe the manuscript would benefit from a more specific title.

Reviewer #3 (Remarks to the Author):

I believe reviewing other work that looks at stable absorption to the CH site (or any site-specific bonding on Si(111) as reviewer 1 said) is appropriate, regardless of the molecule size. It may be that the surface states at the CH site promote bonding, not the 'hole', or it might be that larger molecules partially 'dock' in the CH in the same way. These are very similar problems to reject as outside the scope of review for this paper.

As reviewer one says, site specific bonding on this surface has already been observed. So despite the rebuttal, I still think the claim “this atomic behavior somehow has been unnoticed and neglected” is too strong and unnecessary.

On title, the authors have shown that atoms stably bond to the CH site of Si(111)-7x7. To say they love all holes might be more appropriate for a review paper considering a raft of work where across different surfaces with holes atoms stably bond to all the holes. But I leave this for the editor to consider.

Note, when reviewer one asked for the STM tunnelling settings to be included, they did not mean in the response, they meant in the figure captions of the paper.

When reviewer 1 asks how you know only one Ag atom is involved, the authors' answer is not convincing. They say they observed no contamination, which confirms that this is likely Ag, but it doesn't preclude two Ag atoms being required to cause the brightness change observed. Maybe there are single Ag atoms in the other CHs and it's only when a second is added that the brightness change is observed. I think the authors are probably correct, but a discussion of how they are certain it is a single atom process would help address this.

Reviewer #4 (Remarks to the Author):

I have read the response letter of the authors to the original reviewer #2 as well as the revised manuscript. I believe that the authors have done a fair amount to address the reviewer's concerns related to the DFT calculations, and I find that the additional calculations performed by the authors at least partially address the reviewer's comments. At this point, the DFT calculations indirectly support the experimental results, and perhaps the authors would consider this sufficient. However, I am surprised that they did not attempt to compute the STM images, themselves using their DFT setup, as this would really have been the connection between theory and experiment that the reviewer is probably seeking. For the single Ag atom on the Si(111)-7x2 or Si(111)-5x5 surfaces, I expect that the usual approximations of Bardeen [see J. Bardeen Phys. rev. Lett. 6, 57 (1961) and Hofer et al. Rev. Mod. Phys. 75, 1287 (2003)] or Tersoff-Hamann [see Tersoff and Hamann Phys. Rev. B 31, 805 (1985)] would have been sufficient, although some caveats were pointed out by Hayes and Tuckerman [J. Phys. Chem. C 114, 15102 (2010)] for certain adsorbates on Si surfaces. The perturbative methods referred to above are implemented in the Quantum Espresso code if the authors would like to give it a try. I am only suggesting this as something that would likely increase

the quality of the paper, but it really should be up to the authors to decide if they want to pursue this.

Response to the reviewers

We thank all the reviewers for their insightful comments. In the following sections we present our response to the reviewers on a point-by-point basis:

Reviewer #1 (Remarks to the Author):

The authors have answered most of my original concerns and have improved the paper by adding additional simulation results. However, there are a few points that I would still like to see better addressed by the authors:

1. Effect of the tip: While I accept the arguments provided in the response by the authors that the tunnelling electrons should not affect the dynamics of the adsorbed Ag atoms at the scanning bias, it would be interesting to see whether or not the continuous scanning affects the stability of the Ag atoms in CHs at room temperature. Is it possible that the presence of the tip above the scanning area somehow aids the stability of Ag atoms in CHs, either through an electric field mechanism, or an effect similar to the 'tip shading'? It would be useful to see statistical measurements of the sample surface occupancy with Ag atoms taken at significant time intervals (hours, days) without continuous scanning in order to completely disentangle any effect that the tip may have.

Our response: We thank the reviewer for his comment and based on the suggestions we have now performed a set of new experiments and we present the details below:

In these experiments we considered an area of 100x100 nm² with 116 Ag atoms inside the CHs. During the experiment the tip was retracted by 1 μm, the tip contact was reestablished. The same area of the surface was scanned again and checked if any of the Ag atoms inside CHs that were present changed the adsorption site and/or escaped from the CH. NONE of the 116 Ag atoms escaped from any of the CHs at RT.

This experiment clearly show that the tip influence is negligible.

The data that is related to the above experiment is now available in the supplementary material and there are few changes in the main text as well.

Further to the above comment, the reviewer also suggested new experiments on the influence of the tip and tunneling conditions on the stability of Ag inside CH is interesting but we believe this is really out of the scope of this paper and we hope to address this in the future experiments.

2. Many of the figures in the main text and in the supplementary material are still missing image sizes and scanning parameters.

Our response: We apologize for omitting this somehow and we believe this very important information on the figures, which has been now corrected.

3. The added excerpt from a thesis in the supplementary material lacks any information about how the data was obtained and how many experiments were performed.

Our response: We thank the reviewer for pointing this out, but the experimental details have been already provided in the methods section, which are same as the one, which the first author reported in his PhD thesis. The reference to the thesis was given to make sure that readers are aware that the original experiments were done as early as 2005.

4. As reviewer 3 pointed out, the title is still too general. Even though the added simulations suggest that there might be more atoms that behave in the same way as Ag, it is certainly not the case for O atoms for example. I believe the manuscript would benefit from a more specific title.

Our response: We agree with the reviewer and we have now changed the title of the manuscript to

“Periodic corner holes on the Si(111)-7x7 surface can trap silver atoms”

Response to the reviewers

Reviewer #3 (Remarks to the Author):

I believe reviewing other work that looks at stable absorption to the CH site (or any site-specific bonding on Si(111) as reviewer 1 said) is appropriate, regardless of the molecule size. It may be that the surface states at the CH site promote bonding, not the 'hole', or it might be that larger molecules partially 'dock' in the CH in the same way. These are very similar problems to reject as outside the scope of review for this paper.

Our response: In our paper we clearly showed that the Ag atom is forming a chemical bond with the Si atom in the CH, which is not on the surface but inside the CH giving the Ag atoms a higher stability. In this manuscript we are addressing the atomic adsorption and/or their entrapment, therefore, molecular adsorption is not discussed.

As reviewer one says, site specific bonding on this surface has already been observed. So despite the rebuttal, I still think the claim "this atomic behavior somehow has been unnoticed and neglected" is too strong and unnecessary.

Our response: As the reviewer suggested we already removed the phrase "unnoticed and neglected" in the previous response to the reviewers.

Here is the sentence from the manuscript: "Despite a broad interest on the Si(111) reconstructed surface for some reason chemical characteristics of one adsorption site i.e. corner hole has not been explored enough."

On title, the authors have shown that atoms stably bond to the CH site of Si(111)-7x7. To say they love all holes might be more appropriate for a review paper considering a raft of work where across different surfaces with holes atoms stably bond to all the holes. But I leave this for the editor to consider.

Our response: We have changed the title of the paper as suggested by reviewer 1.

Note, when reviewer one asked for the STM tunnelling settings to be included, they did not mean in the response, they meant in the figure captions of the paper.

Our response: We have now addressed this issue.

When reviewer 1 asks how you know only one Ag atom is involved, the authors' answer is not convincing. They say they observed no contamination, which confirms that this is likely Ag, but it doesn't preclude two Ag atoms being required to cause the brightness change observed. Maybe there are single Ag atoms in the other CHs and it's only when a second is added that the brightness change is observed. I think the authors are probably correct, but a discussion of how they are certain it is a single atom process would help address this.

Our response: We thank the reviewer for this comment, and we have now addressed this issue once again as follows:

We believe, the number Ag atoms adsorbed on the CHs is only one the reason being, (i) The Si(111)-7x7 surface was prepared clean

(ii) On the clean Si(111)7x7 surface, no bright features like Ag/Si(111)-7x7 system were visible at all.

(iii) The amount of the Ag atoms was very little.

(iv) A careful comparison of the clean and Ag adsorbed Si(111)-7x7 surface showed increase of the brightness around CH, which is related to the jump made by 1 Ag-atom to the CH.

(iv) During this process, we did not observe brightness of two different kinds. The brightness of the CHs throughout the area under investigation was consistent, which provides the proof that the number Ag atoms in these CHs were the same.

Response to the reviewers

(v) In the STM image we did not observe any other brighter sites other than those with 1 Ag inside the CH, which would suggest the presence of more than one Ag atom.

(vi) we have analysed the STM images with Ag on the Si(111)-7x7 surface and plotted the distribution of the height differences between corresponding adatoms next to corner holes with and without the Ag atoms in the FHUCs and UHUCs. The height distributions follow a normal Gauss distribution indicating existence of one height value for Ag with a standard deviation related to measurement error. The data related to the above experiment is now also available in the supplementary material.

(vii) Finally, the latest DFT calculations confirms these experimental observations i.e., increased brightness around the CH with 1 Ag atom inside.

Reviewer #4 (Remarks to the Author):

I have read the response letter of the authors to the original reviewer #2 as well as the revised manuscript. I believe that the authors have done a fair amount to address the reviewer's concerns related to the DFT calculations, and I find that the additional calculations performed by the authors at least partially address the reviewer's comments. At this point, the DFT calculations indirectly support the experimental results, and perhaps the authors would consider this sufficient. However, I am surprised that they did not attempt to compute the STM images, themselves using their DFT setup, as this would really have been the connection between theory and experiment that the reviewer is probably seeking. For the single Ag atom on the Si(111)-7x2 or Si(111)-5x5 surfaces, I expect that the usual approximations of Bardeen [see J. Bardeen Phys. rev. Lett. 6, 57 (1961) and Hofer et al. Rev. Mod. Phys. 75, 1287 (2003)] or Tersoff-Hamann [see Tersoff and Hamann Phys. Rev. B 31, 805(1985)] would have been sufficient, although some caveats were pointed out by Hayes and Tuckerman [J. Phys. Chem. C 114, 15102 (2010)] for certain adsorbates on Si surfaces. The perturbative methods referred to above are implemented in the Quantum Espresso code if the authors would like to give it a try. I am only suggesting this as something that would likely increase the quality of the paper, but it really should be up to the authors to decide if they want to pursue this.

Our response: We thank the reviewer for his suggestion. In the manuscript we have now included the STM images along with the above-mentioned references.

REVIEWERS' COMMENTS

Reviewer #1 (Remarks to the Author):

The authors have now answered most of my concerns very comprehensively, as well as those of the other referees. I believe that the paper is much better for the additional experimental and simulated results. I therefore recommend the paper in its current form for publication in Nature Communications.

Reviewer #3 (Remarks to the Author):

I would like to thank the reviewers for their changes to the manuscript, additional work and supplementary information which I think now answers all outstanding points. I am happy to recommend acceptance.

Reviewer #4 (Remarks to the Author):

I am pleased to see that the authors included the simulated STM images in Figure 8 of the revised manuscript. I think this modified version of the paper is essentially ready to be published. However, before the authors finalize their manuscript, they should indicate in Figure 8 where the Ag atoms are (perhaps indicating in a manner similar to Figure 1). They should also use the same scale in Figure 8(b) as they do in 8(a) so that the changes relative to the pristine surface can be easily observed. I will leave this up to the authors to fix. There is no need to see the manuscript again.

Response to the reviewers

We thank all the reviewers for their positive comments. In the following sections we present our response to the reviewers on a point-by-point basis:

Reviewer #1 (Remarks to the Author):

Comment: The authors have now answered most of my concerns very comprehensively, as well as those of the other referees. I believe that the paper is much better for the additional experimental and simulated results. I therefore recommend the paper in its current form for publication in Nature Communications.

Our response: We thank the reviewer for the positive recommendation.

Reviewer #3 (Remarks to the Author):

Comment: I would like to thank the reviewers for their changes to the manuscript, additional work and supplementary information which I think now answers all outstanding points. I am happy to recommend acceptance.

Our response: We thank the reviewer for the positive recommendation.

Reviewer #4 (Remarks to the Author):

Comment: "I am pleased to see that the authors included the simulated STM images in Figure 8 of the revised manuscript. I think this modified version of the paper is essentially ready to be published. However, before the authors finalize their manuscript, they should indicate in Figure 8 where the Ag atoms are (perhaps indicating in a manner similar to Figure 1). They should also use the same scale in Figure 8(b) as they do in 8(a) so that the changes relative to the pristine surface can be easily observed. I will leave this up to the authors to fix. There is no need to see the manuscript again."

Our response: We thank the reviewer for the positive recommendation and the suggestions based on which we have now updated Figure 8. In the revised figure we have used the same scales for Figure 8 (a) and (b). Additionally, we have also indicated the corner holes with the Ag atoms like Figure 1.